# Uncertainty Estimation with Recursive Feature Machines

**Daniel Gedon**[*1]  **Amirhesam Abedsoltan**[*2]  **Thomas B. Schön**[1]  **Mikhail Belkin**[2,3]

[1]Department of Information Technology, Uppsala University, Sweden
[2]Department of Computer Science and Engineering, UC San Diego, USA
[3]Halıcıoğlu Data Science Institute, UC San Diego, USA

## Abstract

In conventional regression analysis, predictions are typically represented as point estimates derived from covariates. The Gaussian Process (GP) offer a kernel-based framework that predicts and quantifies associated uncertainties. However, kernel-based methods often underperform ensemble-based decision tree approaches in regression tasks involving tabular and categorical data. Recently, Recursive Feature Machines (RFMs) were proposed as a novel feature-learning kernel which strengthens the capabilities of kernel machines. In this study, we harness the power of these RFMs in a probabilistic GP-based approach to enhance uncertainty estimation through feature extraction within kernel methods. We employ this learned kernel for in-depth uncertainty analysis. On tabular datasets, our RFM-based method surpasses other leading uncertainty estimation techniques, including NGBoost and CatBoost-ensemble. Additionally, when assessing out-of-distribution performance, we found that boosting-based methods are surpassed by our RFM-based approach.

## 1 INTRODUCTION

Regression analysis traditionally predicts future outcomes by providing definitive values based on empirical data. However, as the applications of predictive modelling expand into critical areas like healthcare [Nicora et al., 2022, Tran et al., 2021, Avati et al., 2018] and weather forecasting [Gneiting and Katzfuss, 2014], there is an increasing need to understand the confidence or uncertainty surrounding these predictions, beyond just point estimates. As stated in Kompa et al. [2021] "*medical ML should have the ability to say "I don't know" and potentially abstain from providing a diagnosis or prediction when there is a large amount of uncertainty for a given patient*". A rising number of publications underscore the importance of uncertainty quantification, evident in fields like radiology [Chua et al., 2023], digital pathology [Linmans et al., 2023], cancer digital histopathology [Dolezal et al., 2022], and radiation oncology [Barragán-Montero et al., 2022], to name a few.

The Recursive Feature Machine (RFM) [Radhakrishnan et al., 2024a] represents an innovative data-adaptive kernel-based method, which provides a unique lens for data interpretation. Our research explores the capabilities of RFMs, focusing on their aptitude for uncertainty estimation in both in-distribution and out-of-distribution contexts. We pit our probabilistic RFMs against other prominent techniques, especially state-of-the-art probabilistic decision tree-based methods like NGBoost [Duan et al., 2020] and CatBoost-ensembles [Prokhorenkova et al., 2018], underscoring their competitive edge.

The Gaussian process (GP) is often the method of choice for estimating uncertainty in predictions [Rasmussen and Williams, 2006], offering a sophisticated perspective beyond point estimates. However, with the ongoing evolution in machine learning, decision tree-based techniques such as NGBoost and CatBoost-ensembles are gaining traction. These methods not only challenge the GP in terms of prediction accuracy but have also showcased superior results in specific uncertainty metrics like Negative Log Likelihood (NLL), coverage error (CE) and prediction interval length (IL), especially for tabular or categorical data.

In our study, we demonstrate that by combining GPs with the data-adaptive kernel derived from the RFM, we can bridge this performance gap, achieving results that are on par with or even surpass gradient-based boosting approaches. In summary, (i) we introduced the RFM to the GP community and (ii) established that the performance of RFM is comparable to, or even superior to, existing state-of-the-art methods. More specifically, we have the following contributions:

- Our findings reveal that GP-RFM is a strong alternative

---
[*]Equal contribution.

to leading boosting-based techniques, particularly by enhancing uncertainty estimation in tabular datasets via features generated from the RFM. This capability to match or in certain instances exceed the performance of existing top-tier methods establishes the RFM as a new benchmark for applications that demand accurate uncertainty assessments.

- We bring RFMs to the GP community and illustrate that features derived from the RFM notably improve uncertainty performance on tabular datasets. Comparing the RFM with traditional GP techniques, we further show that the RFM can extract more general feature representations due to its ability to capture correlation between features. This can in turn significantly improve the resulting uncertainty estimates.

- To highlight the robustness of the RFM we compare it on out-of-distribution data for label and covariate shift where the RFM surpasses other uncertainty quantification methods.

## 2 PRIOR WORK

Numerous uncertainty quantification methods have been proposed in the literature for utilization with tabular data. Here, we focus on discussing flexible methods with state-of-the-art predictive performance.

**Gaussian processes.** As a non-parametric, flexible Bayesian regression model, the Gaussian process is a well-studied and natural choice for uncertainty quantification [Rasmussen and Williams, 2006]. The GP is characterized by a mean function and a kernel function as covariance. The crucial challenge is to choose the right kernel as it encodes high-level assumptions about the data. Commonly, the Radial Basis Function (RBF) or Laplace kernel is chosen, which has a limited number of parameters to optimize. For more flexibility, kernels with Automatic Relevance Determination (ARD) introduce covariate weighting through learnable parameters [MacKay, 1992, Neal, 1996]. Vivarelli and Williams [1998] generalizes the diagonal ARD weighting to general positive-definite weighting matrices or low-rank factorisations. Garnett et al. [2014] and Letham et al. [2020] use a factorized weighting matrix and approximate the posterior with Laplace approximation for active learning and Bayesian optimization. For the latter, sparse axis-aligned Subspace GPs leverage structural sparsity in the kernel [Eriksson and Jankowiak, 2021]. Instead of utilizing advanced kernels, we can equivalently transform the input and use standard kernels [MacKay et al., 1998]. Neural networks have been studied as feature extractors [Calandra et al., 2016, Wilson et al., 2016], or where the last layer approximates a GP [Huang et al., 2015, Liu et al., 2020]. Our approach combines both strategies, leveraging the recently proposed Recursive Feature Machine [Radhakrishnan et al., 2024a],

which introduces a novel feature-extracting kernel with the probabilistic expressivity of GPs.

**Probabilistic boosting.** Boosting-based approaches [Freund and Schapire, 1995, Friedman, 2001] allow for flexible models, which have found widespread application on tabular datasets [Shwartz-Ziv and Armon, 2022, Grinsztajn et al., 2022, McElfresh et al., 2023]. Such methods include among others AdaBoost, XGBoost, LightGBM or CatBoost [Chen and Guestrin, 2016, Ke et al., 2017, Prokhorenkova et al., 2018]. For classification problems, most methods have a natural probabilistic interpretation through estimated class probabilities. However, for regression problems, there is no such straightforward concept. Therefore, probabilistic extensions of boosting such as NGBoost, CatBoost-Ensembles [Duan et al., 2020, Malinin et al., 2021] or extensions to Random Forests [Schlosser et al., 2019, Shaker and Hüllermeier, 2020] have been proposed. Notably, when comparing the performance of probabilistic boosting approaches against our GP-RFM, our approach performs on par or even outperforms them across a range of evaluation metrics and tabular regression datasets.

**Neural networks.** The ability to learn features from data is a key advantage of the predictive power of neural networks (NN). For uncertainty quantification, Bayesian NNs [MacKay, 1992, Neal, 1996] are a natural choice. However, the need for approximate inference methods such as variational inference [Graves, 2011, Blundell et al., 2015] or Markov Chain Monte Carlo [Welling and Teh, 2011] makes them computationally expensive. Conversely, the use of Monte Carlo dropout [Gal and Ghahramani, 2016] provides less reliable uncertainty estimates [Ovadia et al., 2019, Gustafsson et al., 2020] than ensembles of NNs [Lakshminarayanan et al., 2017]. Although deep ensembles set the gold standard for NNs, they necessitate training multiple NNs resulting in high computational and memory burden. We leverage the idea of feature learning in NN through the use of RFMs since the learnt features in the latter are intricately linked to features learnt in feedforward NNs [Radhakrishnan et al., 2024a].

## 3 BACKGROUND

Most machine learning algorithms focus on estimating the predictive model $f(\boldsymbol{x}) = \mathbb{E}[y|\boldsymbol{x}]$ from a training dataset $\mathcal{D} = (\boldsymbol{X}, \boldsymbol{y}) = \{\boldsymbol{x}_i \in \mathbb{R}^d, y_i \in \mathbb{R}\}_{i=1}^n$. However, in many applications, this is not sufficient. We are therefore interested in augmenting point estimates with reliable uncertainty quantification to obtain the predictive distribution $p(f(\boldsymbol{x}_*)|\boldsymbol{x}_*, \mathcal{D})$ for a new test data point $\boldsymbol{x}_*$. In our approach, we leverage GPs in conjunction with feature learning kernels through RFMs.

## 3.1 KERNEL MACHINES

Kernel machines [Schölkopf and Smola, 2002] are non-parametric predictive models. Given training data $\mathcal{D}$ a kernel machine is a model of the form

$$f(\boldsymbol{x}) = \sum_{i=1}^{n} \alpha_i k(\boldsymbol{x}, \boldsymbol{x}_i). \tag{1}$$

Here, $k : \mathbb{R}^d \times \mathbb{R}^d \to \mathbb{R}$ is a positive semi-definite symmetric kernel function [Aronszajn, 1950]. According to the representer theorem [Kimeldorf and Wahba, 1970], the unique solution to the infinite-dimensional optimization problem

$$\arg\min_{f \in \mathcal{H}} \sum_{i=1}^{n} (f(\boldsymbol{x}_i) - y_i)^2 + \lambda \|f\|_{\mathcal{H}}^2 \tag{2}$$

has the form given in (1). Here $\mathcal{H}$ is the (unique) reproducing kernel Hilbert space corresponding to $k$, and $\lambda$ is the ridge regularizer. It can be seen that $\boldsymbol{\alpha} = (\alpha_1, \ldots, \alpha_n)$ in (1) is the unique solution to the linear system,

$$(k(\boldsymbol{X}, \boldsymbol{X}) + \lambda \boldsymbol{I}_n)\,\boldsymbol{\alpha} = \boldsymbol{y}. \tag{3}$$

## 3.2 GAUSSIAN PROCESSES

To extend kernel machines into a probabilistic setting, we can define a distribution over the predictive function which yields a GP $f \sim \mathcal{GP}(m, k)$ specified by its mean function $m$ and its covariance function $k$. Because of its properties, we utilize the kernel function $k$ as the covariance function in the GP. The posterior predictive distribution of the GP is then given by

$$p(f(\boldsymbol{x}_*)|\boldsymbol{x}_*, \mathcal{D}) = \mathcal{N}\left(f(\boldsymbol{x}_*), \boldsymbol{\Sigma}(\boldsymbol{x}_*)]\right), \tag{4}$$

with the mean as in (1) and the covariance $\boldsymbol{\Sigma}(\boldsymbol{x}_*) = \mathbb{V}[f(\boldsymbol{x}_*)] = k(\boldsymbol{x}_*, \boldsymbol{x}_*) - \boldsymbol{k}_*^\top (\boldsymbol{K} + \sigma^2 \boldsymbol{I})^{-1} \boldsymbol{k}_*$. We denote the kernel matrix as $\boldsymbol{K}$ with $K_{i,j} = k(\boldsymbol{x}_i, \boldsymbol{x}_j)$, $\boldsymbol{k}_* = k(\boldsymbol{X}, \boldsymbol{x}_*)$ and the measurement noise variance as $\sigma^2$.

For the mean function, we choose the constant function $m = 0$. The choice of kernel encodes high-level assumptions about the resulting function. We consider an exponential kernel of the form

$$k(\boldsymbol{x}, \boldsymbol{z}) = \exp\left(g(\boldsymbol{x}, \boldsymbol{z})\right). \tag{5}$$

When we define $g(\boldsymbol{x}, \boldsymbol{z}) = -\frac{1}{2\ell^2}\|\boldsymbol{x} - \boldsymbol{z}\|^2$, we arrive at the widely adopted Radial Basis Function (RBF) kernel. Conversely, $g(\boldsymbol{x}, \boldsymbol{z}) = -\frac{1}{\ell}\|\boldsymbol{x} - \boldsymbol{z}\|$ leads to the Laplace kernel.

The parameters $\boldsymbol{\theta}$ of the kernel include the noise variance $\sigma$ and the length scale $\ell$. These parameters are often found by solving an optimization problem dictated by Maximum Likelihood Estimation (MLE). Specifically, we can estimate these parameters in a Bayesian framework by minimizing the Negative Log Likelihood (NLL), defined as $-\log p(\boldsymbol{y}|\boldsymbol{X}, \boldsymbol{\theta})$.

## 3.3 RECURSIVE FEATURE MACHINES

A fundamental limitation of kernel machines is their reliance on kernel functions that are not adaptive to data. As a result, for certain tasks, kernel machines can significantly underperform compared to neural networks. The recently introduced RFM constitute a type of kernel machine capable of learning features, making them data-adaptive.

To develop kernel machines that can learn features, RFM adds a positive semi-definite, symmetric matrix, $\boldsymbol{M}$, as a learnable parameter into the kernel function. Specifically, this is suited for kernel functions that depend on the distance between points, such as $k(\boldsymbol{x}, \boldsymbol{z}) = \phi(\|\boldsymbol{x} - \boldsymbol{z}\|^2)$ where $\phi : \mathbb{R} \to \mathbb{R}$ and $\boldsymbol{x}, \boldsymbol{z} \in \mathbb{R}^d$. We incorporate the learnable matrix $\boldsymbol{M}$ by using the Mahalanobis distance

$$\|\boldsymbol{x} - \boldsymbol{z}\|_{\boldsymbol{M}} := \sqrt{(\boldsymbol{x} - \boldsymbol{z})^T \boldsymbol{M} (\boldsymbol{x} - \boldsymbol{z})}. \tag{6}$$

Therefore, the matrix $\boldsymbol{M}$ re-weights the individual covariates and can incorporate correlation between covariates, for which we call $\boldsymbol{M}$ the *feature matrix*. While any kernel function can be used for $\phi$, we utilize the Laplace kernel based on the Mahalanobis distance

$$k_{\boldsymbol{M}}(\boldsymbol{x}, \boldsymbol{z}) := \exp\left(-\frac{1}{\gamma}\|\boldsymbol{x} - \boldsymbol{z}\|_{\boldsymbol{M}}\right). \tag{7}$$

The prediction function corresponding to this kernel is

$$f_{\boldsymbol{M}}(\boldsymbol{x}) = k_{\boldsymbol{M}}(\boldsymbol{x}, \boldsymbol{X})\boldsymbol{\alpha}, \tag{8}$$

with $\boldsymbol{\alpha} = k_{\boldsymbol{M}}(\boldsymbol{X}, \boldsymbol{X})^{-1}\boldsymbol{y}$. To learn the feature matrix $\boldsymbol{M}$ we make use of the proposed idea of the Average Gradient Outer Product (AGOP) from Radhakrishnan et al. [2024a]: We start by initializing $\boldsymbol{M}^{(0)} = \boldsymbol{I}_d$. At each iteration step $t$, we first solve for the kernel weights $\boldsymbol{\alpha}$ from (8) with fixed $\boldsymbol{M}$. Second, we update $\boldsymbol{M}$ using the AGOP defined as

$$\boldsymbol{M}^{(t+1)} = \frac{1}{n}\sum_{i=1}^{n}\left(\nabla_{\boldsymbol{x}} f_{\boldsymbol{M}^{(t)}}(\boldsymbol{x}_i)\right)\left(\nabla_{\boldsymbol{x}} f_{\boldsymbol{M}^{(t)}}(\boldsymbol{x}_i)\right)^T. \tag{9}$$

Essentially the AGOP and the resulting matrix $\boldsymbol{M}$ is the covariance matrix of the function gradients. Intuitively, RFM prioritises the covariates that have the most impact on the prediction function. Thus, RFMs learn the presentation most relevant to the underlying task.

A special case of RFMs is when we restrict the feature matrix $\boldsymbol{M}$ to be diagonal. This is equivalent to learning a separate length scale for each covariate. In contrast to the *RFM* without this restriction, we refer to the diagonally restricted model as *RFM-diag*.

**Remark.** Another way of covariate weighting specifically in GPs is through the extension with Automatic Relevance Determination (ARD) [Neal, 1996]. The RBF kernel is extended by using $g(\boldsymbol{x}, \boldsymbol{z}) = -\frac{1}{\ell^2}\|\boldsymbol{x} - \boldsymbol{z}\|_{\boldsymbol{M}}^2$ with $\boldsymbol{M}^{-1} =$

diag($[\ell_1^2, \ldots, \ell_d^2]$). While barely utilised in practice, a similar construction can be generated for the Laplace kernel with ARD. This effectively increases the parameter vector $\boldsymbol{\theta}$ learnt through MLE optimization in the GP framework.

# 4 METHOD: GP-RFM

While GPs are powerful non-parametric models which offer uncertainty quantification, they are limited by their reliance on kernel functions that are not adaptive to data. RFMs are a type of kernel machine capable of learning features, making them data-adaptive. We propose to incorporate RFMs into GPs by replacing the kernel function $k(\boldsymbol{x}, \boldsymbol{z})$ with the RFM-based kernel $k_{\boldsymbol{M}}(\boldsymbol{x}, \boldsymbol{z})$. Since we are using the Laplace kernel within the RFM, we refer to the resulting construction as *GP-RFM-Laplace*.

Specifically, we consider a combination of a scale kernel with the RFM-based kernel to obtain $\sigma_f^2 k_{\boldsymbol{M}}(\boldsymbol{x}, \boldsymbol{z})$. Since we are interested in the predictive distribution, we can set the mean function $m$ of the GP to zero. The resulting predictive distribution is then given by (4) where $f(\boldsymbol{x})$ is given by (8). The parameters of the GP are the feature matrix $\boldsymbol{M}$ and the kernel parameters $\boldsymbol{\theta}$ consisting of noise variance $\sigma$, length scale $\ell$ as well as scale $\sigma_f$.

The pseudo-code of GP-RFM can be found in Algorithm 1. It is important to highlight that the RFM algorithm, which identifies the matrix $\boldsymbol{M}$ through the AGOP iteration, see (9), shares the same time and memory complexity as the GP regression algorithm, see Radhakrishnan et al. [2024a]. Consequently, incorporating this additional step does not complicate the overall complexity. For a comparison of the actual running times between our algorithm and other methods, please see Appendix C. In that section, we illustrate the time efficiency advantages of our method compared to boosting-based approaches like NGBoost and CatBoost ensembles.

**Training.** We disentangle the training of the GP-RFM-Laplace into two steps. First, we learn the feature matrix $\boldsymbol{M}$ using the recursive iteration between solving for the kernel weights $\boldsymbol{\alpha}$ for (8) and updating $\boldsymbol{M}$ using the AGOP defined in (9). To learn the kernel weights, we solve the linear system in (3) with a Ridge regularization term for stability to obtain $\boldsymbol{\alpha} = (\boldsymbol{K} + \lambda_\alpha \boldsymbol{I}_n)^{-1} \boldsymbol{y}$. For the AGOP we need to compute the gradient of the prediction function w.r.t. the covariates $\boldsymbol{x}_i$. For the Laplace kernel, there exist closed-form solutions which we make use of [Radhakrishnan et al., 2024a]. Second, we learn the GP-specific kernel parameters $\boldsymbol{\theta}$ by MLE optimization, specifically by minimizing the NLL with fixed $\boldsymbol{M}$.

**Eliminating spurious correlation.** In many datasets, there exist spurious correlations between covariates. To avoid overfitting to spurious covariate correlation, we add a

---

**Algorithm 1** Training of the GP-RFM model

**First Stage: Learning data-adaptive kernel $k_{\boldsymbol{M}}$**

1: **Input:** $\boldsymbol{X}, \boldsymbol{y}, k_{\boldsymbol{M}}, T, \lambda_{\boldsymbol{M}}$
2: **Output:** $\boldsymbol{M}$
3: Initialize $\boldsymbol{M} = \boldsymbol{I}_{d \times d}$          ▷ Identity matrix
4: **for** $t$ in $T$ **do**
5:      Compute $k_{\text{train}} = k_{\boldsymbol{M}}(\boldsymbol{X}, \boldsymbol{X})$
6:      Calculate $\boldsymbol{\alpha} = k_{\text{train}}^{-1} \boldsymbol{y}$    ▷ $f(\boldsymbol{x}) = k_{\boldsymbol{M}}(\boldsymbol{x}, \boldsymbol{X})\boldsymbol{\alpha}$
7:      Update $\boldsymbol{M}$:      ▷ Outer product of gradients
       $\boldsymbol{M} = \frac{1}{n} \sum_{i=1}^{n} (\nabla f(\boldsymbol{x}_i))(\nabla f(\boldsymbol{x}_i))^\top + \lambda_{\boldsymbol{M}} \boldsymbol{I}_d$

**Second Stage: Train the GP with fixed $\boldsymbol{M}$**

8: **Input:** $\boldsymbol{X}, \boldsymbol{y}, \boldsymbol{M}$, possible composite kernel $k$
9: **Output:** $f_{\boldsymbol{M}}, \boldsymbol{\Sigma}_{\boldsymbol{M}}$
10: Define kernel with hyperparameters $k_{gp} = k(k_{\boldsymbol{M}}, \boldsymbol{\theta})$
11: Optimize kernel hyperparameters $\boldsymbol{\theta}$ through MLE
12: Compute predictive quantities $f_{\boldsymbol{M}}(\boldsymbol{x}), \boldsymbol{\Sigma}_{\boldsymbol{M}}(\boldsymbol{x})$

---

Ridge regularization term to the AGOP in (9) to obtain

$$\boldsymbol{M}^{(t+1)} = \frac{1}{n} \sum_{i=1}^{n} \left(\nabla_{\boldsymbol{x}} f_{\boldsymbol{M}^{(t)}}(\boldsymbol{x}_i)\right) \left(\nabla_{\boldsymbol{x}} f_{\boldsymbol{M}^{(t)}}(\boldsymbol{x}_i)\right)^T \\ + \lambda_{\boldsymbol{M}} \boldsymbol{I}_d. \quad (10)$$

The Ridge regularization acts in this case as a noise filter by shrinking off-diagonal elements of $\boldsymbol{M}$ towards zero. Therefore, the learnt feature correlation in the off-diagonal elements of $\boldsymbol{M}$ is only kept if it is supported by the data, making the model more robust to random variations in the data.

**Uncertainty quantification.** During inference, we can quantify the uncertainty of the GP-RFM-Laplace by computing the predictive variance $\mathbb{V}[f(\boldsymbol{x}_*)]$ from (4). In traditional GPs, we have to choose the kernel function carefully to encode assumptions about the resulting function. In contrast, the GP-RFM-Laplace is able to learn features from the data and therefore it is more flexible in its assumptions about the resulting function. While Radhakrishnan et al. [2024a] showed the predictive power of RFMs, we show that the learnt features provide additional insight into the variability or ambiguity of the data which is crucial for uncertainty quantification.

**Implementation details.** We implement the GP-RFM-Laplace in PyTorch [Paszke et al., 2019]. For the computation of the feature matrix in the first step of the training procedure, we rely on the official implementation by Radhakrishnan et al. [2024a]. For the GP implementation, we use GPyTorch [Gardner et al., 2018] which provides a modular implementation of GPs in PyTorch. To optimize the GP parameters, we use the Adam optimizer [Kingma and Ba, 2014] with a cosine annealing learning rate scheduler [Loshchilov and Hutter, 2016]. The hyperparamet-

ers we have to select are the learning rate, the Ridge regularization parameters $\lambda_\alpha$ and $\lambda_M$ for the solver and the AGOP, respectively. Code is openly available at https://github.com/dgedon/rfm_uncertainty.

# 5 EXPERIMENTS

**Datasets.** We evaluate our GP-RFM-Laplace on a variety of regression tasks. Specifically, we use two tabular regression benchmarks with datasets from UCI [Asuncion and Newman, 2007] and OpenML [Vanschoren et al., 2014], respectively. For the UCI benchmark we use 7 datasets inspired by Duan et al. [2020] and for the OpenML benchmark, we utilize the collection of 16 numerical regression datasets by Grinsztajn et al. [2022].

**Hyperparameter tuning** We follow the protocol proposed in Hernández-Lobato and Adams [2015] for data splitting and hyperparameter tuning. For the UCI benchmark, we follow Duan et al. [2020] to hold out 10% of the data as a test set. For the OpenML benchmark, we follow Grinsztajn et al. [2022] to hold out 30% of the data as a test set. The remaining data is split into a 70% training set and a 30% validation set to tune the hyperparameters. We use grid-search over all combinations of hyperparameters and select the best hyperparameters based on the validation set NLL. Details on the hyperparameter search space can be found in the appendix. Finally, we train the model on the full training set and evaluate it on the test set. The process is repeated for 20 random seeds and we report the mean and standard deviation of the results.

**Baselines.** We compare our GP-RFM-Laplace and its diagonal version GP-RFM-Laplace-diag to a variety of probabilistic baseline methods. The details are described in Appendix A. For GPs, we consider the standard *RBF* and *Laplace* kernel. As a neural networks-based GP, we regard *deep kernel learning* [Wilson et al., 2016]. Additionally, we compare our method to kernels with ARD, specifically the *ARD-RBF* [Neal, 1996] which is used in many settings and to the *ARD-Laplace* kernel. The latter is a rarely used kernel in GPs but is a natural extension of the Laplace kernel to incorporate covariate weighting, learnt through MLE optimization. Finally, we use *ARD-Laplace-full* as a direct counterpart to the RFM-Laplace with full weighting matrix but learnt through MLE here instead of AGOP [Vivarelli and Williams, 1998].

Furthermore, we consider probabilistic extensions of boosting approaches, which are known to be powerful for predictive tasks. Firstly, we use *NGBoost* [Duan et al., 2020] which learns the parameters of a Gaussian distribution through boosting enhanced with a natural gradient update. Secondly, we use *CatBoost-Ensemble* [Malinin et al., 2021] for which we use an ensemble of 10 gradient boosting-based models.

From the ensemble, the predictive distribution is obtained by computing statistics of the individual predictions. Following Duan et al. [2020], we standardize covariates and labels to have zero mean and unit variance for all GP-based methods but not for the boosting-based methods.

**Evaluation metrics.** We are interested in the predictive performance of the models as well as their uncertainty quantification. Therefore, we evaluate the models on their *root mean squared error (RMSE)* as well as their *NLL* on the test set. We also require the model uncertainty to be calibrated, i.e. the predictive distribution should reflect the likelihood of prediction errors. To evaluate calibration, we compute the *95% coverage error (CE)* which refers to the proportion of data points for which the 95% prediction interval does not contain the true value. For the model to be well-calibrated, the coverage should be 95% and the corresponding CE should be zero. Finally, we evaluate the *interval length (IL)* of the 95% confidence interval. This measure is important for models with similar CE since a smaller IL indicates a more precise uncertainty quantification.

## 5.1 MAIN RESULTS

Here we present the main results of our experiments. We compare our GP-RFM-Laplace to all baseline methods on the UCI and OpenML benchmark datasets. Due to varying scales, we normalize metrics for comparison across datasets. We achieve this by calculating the minimum and maximum values for each dataset across all methods and seeds, followed by normalizing results to the range $[0, 1]$. The results for each dataset of the OpenML benchmark in terms of NLL and RMSE are in Tables 1 and 2, respectively. Summary figures for NLL, RMSE and CE are shown in Figure 1 using violin plots to indicate the distribution of the results including a boxplot for the median and quartiles. The results for the UCI benchmark are shown in Appendix B, Figure 6. Note that the results for IL are omitted from the summary figure as comparing IL across datasets is not meaningful. Detailed performance results for each method on all datasets individually can be found in the Appendix C.

We observe that both GP-RFM-Laplace variants are only outperformed by the CatBoost-Ensemble in terms of NLL. However, the GP-RFM-Laplace is the best method in terms of RMSE, closely followed by the GP-ARD-Laplace. Regarding calibration in terms of CE, we observe that the boosting methods are dominant, followed by the GP-RFM-Laplace. Overall, both the GP-RFM-Laplace and the GP-ARD-Laplace perform similarly well across all metrics, demonstrating a competitive approach to boosting-based approaches for probabilistic regression.

Notably, the ARD-Laplace-full, serving as a complement to the RFM-Laplace, exhibits a significantly poorer performance while both methods utilize full feature matrices $M$.

Table 1: Probabilistic performance measured by *NLL* ($\downarrow$) on the OpenML benchmark. The best method for each dataset by the mean value is bolded; the second best is underlined. Detailed results for all metrics are in Appendix C.1.

| | Gaussian Process | | | | | | Ours | | Boosting | |
| Dataset | RBF | Laplace | deep KL | ARD RBF | Lap. | Lap.-full | RFM | RFM-diag | NGBoost | CatBoost |
|---|---|---|---|---|---|---|---|---|---|---|
| cpu-act | 2.80 $_{\pm0.07}$ | 2.55 $_{\pm0.02}$ | 2.67 $_{\pm0.03}$ | 2.67 $_{\pm0.04}$ | 2.30 $_{\pm0.02}$ | 3.71 $_{\pm0.10}$ | 2.21 $_{\pm0.01}$ | **2.17** $_{\pm0.01}$ | 2.33 $_{\pm0.14}$ | **2.17** $_{\pm0.03}$ |
| pol | 3.73 $_{\pm0.01}$ | 3.43 $_{\pm0.01}$ | 3.40 $_{\pm0.01}$ | 3.07 $_{\pm0.01}$ | 2.84 $_{\pm0.01}$ | 4.41 $_{\pm0.02}$ | 2.73 $_{\pm0.01}$ | 3.10 $_{\pm0.01}$ | 3.55 $_{\pm0.01}$ | **2.09** $_{\pm0.03}$ |
| elevators | -4.46 $_{\pm0.03}$ | -4.67 $_{\pm0.01}$ | -4.85 $_{\pm0.01}$ | -4.53 $_{\pm0.02}$ | -4.75 $_{\pm0.01}$ | -4.31 $_{\pm0.16}$ | **-4.86** $_{\pm0.01}$ | -4.79 $_{\pm0.01}$ | -4.48 $_{\pm0.02}$ | -4.73 $_{\pm0.02}$ |
| isolet | 3.43 $_{\pm0.01}$ | 3.43 $_{\pm0.01}$ | 2.62 $_{\pm0.09}$ | 3.43 $_{\pm0.01}$ | 3.43 $_{\pm0.01}$ | 3.43 $_{\pm0.01}$ | **2.34** $_{\pm0.04}$ | 2.57 $_{\pm0.02}$ | 2.71 $_{\pm0.02}$ | 2.52 $_{\pm0.02}$ |
| wine | 1.04 $_{\pm0.02}$ | **0.95** $_{\pm0.02}$ | 1.01 $_{\pm0.02}$ | 1.04 $_{\pm0.03}$ | **0.95** $_{\pm0.02}$ | **0.95** $_{\pm0.02}$ | **0.95** $_{\pm0.02}$ | **0.95** $_{\pm0.02}$ | 1.04 $_{\pm0.03}$ | 1.03 $_{\pm0.03}$ |
| Ailerons | -7.18 $_{\pm0.03}$ | -7.31 $_{\pm0.01}$ | -7.31 $_{\pm0.02}$ | -7.19 $_{\pm0.01}$ | -7.33 $_{\pm0.01}$ | -6.72 $_{\pm0.00}$ | -7.37 $_{\pm0.01}$ | -7.36 $_{\pm0.01}$ | **-7.42** $_{\pm0.01}$ | -7.41 $_{\pm0.01}$ |
| houses | 0.16 $_{\pm0.01}$ | 0.07 $_{\pm0.01}$ | 0.09 $_{\pm0.02}$ | 0.16 $_{\pm0.01}$ | -0.10 $_{\pm0.01}$ | 0.17 $_{\pm0.00}$ | -0.07 $_{\pm0.01}$ | -0.04 $_{\pm0.01}$ | 0.07 $_{\pm0.01}$ | **-0.12** $_{\pm0.02}$ |
| houses-16H | 0.84 $_{\pm0.03}$ | 0.72 $_{\pm0.02}$ | 0.95 $_{\pm0.05}$ | 0.84 $_{\pm0.02}$ | 0.72 $_{\pm0.02}$ | 0.90 $_{\pm0.03}$ | 0.69 $_{\pm0.02}$ | 0.71 $_{\pm0.03}$ | 0.57 $_{\pm0.05}$ | **0.51** $_{\pm0.6}$ |
| Bra-houses | -0.99 $_{\pm0.67}$ | -1.49 $_{\pm0.07}$ | -0.64 $_{\pm0.04}$ | -2.19 $_{\pm0.03}$ | -1.82 $_{\pm0.13}$ | 0.27 $_{\pm0.04}$ | -2.11 $_{\pm0.06}$ | -2.07 $_{\pm0.07}$ | -2.18 $_{\pm0.15}$ | **-2.66** $_{\pm0.23}$ |
| bike | 6.17 $_{\pm0.01}$ | 6.15 $_{\pm0.01}$ | 6.08 $_{\pm0.05}$ | 6.05 $_{\pm0.01}$ | 6.03 $_{\pm0.01}$ | 6.07 $_{\pm0.01}$ | 6.04 $_{\pm0.01}$ | 6.03 $_{\pm0.01}$ | 5.62 $_{\pm0.01}$ | **5.58** $_{\pm0.01}$ |
| house-sales | 0.04 $_{\pm0.02}$ | -0.19 $_{\pm0.01}$ | -0.22 $_{\pm0.02}$ | 0.00 $_{\pm0.01}$ | -0.30 $_{\pm0.01}$ | -0.06 $_{\pm0.16}$ | **-0.32** $_{\pm0.01}$ | **-0.32** $_{\pm0.01}$ | -0.27 $_{\pm0.01}$ | -0.31 $_{\pm0.01}$ |
| sulfur | -2.42 $_{\pm0.02}$ | -2.83 $_{\pm0.03}$ | -2.44 $_{\pm0.24}$ | -2.42 $_{\pm0.02}$ | -2.83 $_{\pm0.04}$ | -2.63 $_{\pm0.15}$ | -2.80 $_{\pm0.05}$ | -2.74 $_{\pm0.06}$ | -2.59 $_{\pm0.41}$ | **-2.88** $_{\pm0.08}$ |
| Miami2016 | 0.01 $_{\pm0.00}$ | -0.36 $_{\pm0.01}$ | -0.35 $_{\pm0.02}$ | 0.00 $_{\pm0.00}$ | -0.46 $_{\pm0.01}$ | -0.17 $_{\pm0.18}$ | -0.47 $_{\pm0.01}$ | -0.50 $_{\pm0.01}$ | -0.38 $_{\pm0.01}$ | **-0.53** $_{\pm0.01}$ |
| superconduct | 4.21 $_{\pm0.00}$ | 4.01 $_{\pm0.01}$ | 4.10 $_{\pm0.03}$ | 4.20 $_{\pm0.00}$ | 3.96 $_{\pm0.01}$ | 4.34 $_{\pm0.18}$ | 4.05 $_{\pm0.11}$ | 4.03 $_{\pm0.01}$ | 3.65 $_{\pm0.02}$ | **3.46** $_{\pm0.11}$ |
| california | -0.31 $_{\pm0.01}$ | -0.40 $_{\pm0.01}$ | -0.34 $_{\pm0.09}$ | -0.32 $_{\pm0.01}$ | **-0.64** $_{\pm0.01}$ | -0.40 $_{\pm0.03}$ | -0.60 $_{\pm0.01}$ | -0.61 $_{\pm0.01}$ | -0.45 $_{\pm0.01}$ | -0.59 $_{\pm0.02}$ |
| fifa | 1.27 $_{\pm0.01}$ | 1.24 $_{\pm0.01}$ | 1.19 $_{\pm0.01}$ | 1.23 $_{\pm0.01}$ | 1.22 $_{\pm0.01}$ | 1.19 $_{\pm0.01}$ | 1.21 $_{\pm0.02}$ | 1.19 $_{\pm0.01}$ | **1.09** $_{\pm0.01}$ | 1.10 $_{\pm0.02}$ |

Table 2: Predictive performance measured by *RMSE* ($\downarrow$) on the OpenML benchmark. The best method for each dataset by the mean value is bolded; the second best is underlined. Detailed results for all metrics are in Appendix C.1.

| | Gaussian Process | | | | | | Ours | | Boosting | |
| Dataset | RBF | Laplace | deep KL | ARD RBF | Lap. | Lap.-full | RFM | RFM-diag | NGBoost | CatBoost |
|---|---|---|---|---|---|---|---|---|---|---|
| cpu-act | 8.17 $_{\pm0.22}$ | 6.41 $_{\pm0.10}$ | 3.39 $_{\pm0.30}$ | **3.28** $_{\pm0.09}$ | 3.51 $_{\pm0.11}$ | 6.35 $_{\pm0.23}$ | 3.37 $_{\pm0.15}$ | 4.75 $_{\pm0.15}$ | 12.02 $_{\pm0.23}$ | 4.91 $_{\pm0.14}$ |
| pol | 4.20 $_{\pm0.37}$ | 3.89 $_{\pm0.29}$ | 2.50 $_{\pm0.08}$ | 3.38 $_{\pm0.29}$ | 2.74 $_{\pm0.15}$ | 7.12 $_{\pm1.08}$ | 2.33 $_{\pm0.11}$ | **2.16** $_{\pm0.04}$ | 2.48 $_{\pm0.09}$ | 2.50 $_{\pm0.13}$ |
| elevators $_{(10^{-2})}$ | 0.22 $_{\pm0.00}$ | 0.22 $_{\pm0.00}$ | **0.19** $_{\pm0.00}$ | 0.22 $_{\pm0.01}$ | 0.21 $_{\pm0.00}$ | 0.26 $_{\pm0.02}$ | **0.19** $_{\pm0.00}$ | 0.20 $_{\pm0.00}$ | 0.36 $_{\pm0.01}$ | 0.23 $_{\pm0.00}$ |
| isolet | 7.50 $_{\pm0.05}$ | 7.50 $_{\pm0.05}$ | 3.15 $_{\pm0.24}$ | 7.50 $_{\pm0.05}$ | 7.50 $_{\pm0.05}$ | 7.50 $_{\pm0.05}$ | **2.57** $_{\pm0.10}$ | 3.18 $_{\pm0.08}$ | 4.13 $_{\pm0.07}$ | 3.49 $_{\pm0.07}$ |
| wine | 0.67 $_{\pm0.02}$ | **0.61** $_{\pm0.02}$ | 0.70 $_{\pm0.01}$ | 0.67 $_{\pm0.02}$ | **0.61** $_{\pm0.02}$ | **0.61** $_{\pm0.02}$ | **0.61** $_{\pm0.02}$ | **0.61** $_{\pm0.02}$ | 0.70 $_{\pm0.02}$ | 0.69 $_{\pm0.02}$ |
| Ailerons $_{(10^{-2})}$ | 0.02 $_{\pm0.00}$ | 0.02 $_{\pm0.00}$ | 0.02 $_{\pm0.00}$ | 0.02 $_{\pm0.00}$ | 0.02 $_{\pm0.00}$ | 0.02 $_{\pm0.00}$ | 0.02 $_{\pm0.00}$ | 0.02 $_{\pm0.00}$ | 0.02 $_{\pm0.00}$ | 0.02 $_{\pm0.00}$ |
| houses | 0.26 $_{\pm0.00}$ | 0.25 $_{\pm0.00}$ | 0.26 $_{\pm0.01}$ | 0.24 $_{\pm0.00}$ | **0.21** $_{\pm0.00}$ | 0.25 $_{\pm0.00}$ | 0.22 $_{\pm0.00}$ | 0.22 $_{\pm0.00}$ | 0.28 $_{\pm0.00}$ | 0.23 $_{\pm0.00}$ |
| houses-16H | 0.63 $_{\pm0.03}$ | 0.61 $_{\pm0.03}$ | 0.66 $_{\pm0.06}$ | 0.64 $_{\pm0.03}$ | 0.61 $_{\pm0.03}$ | 0.62 $_{\pm0.03}$ | 0.61 $_{\pm0.03}$ | 0.62 $_{\pm0.03}$ | **0.60** $_{\pm0.03}$ | **0.60** $_{\pm0.03}$ |
| Bra-houses | 0.10 $_{\pm0.04}$ | 0.06 $_{\pm0.03}$ | 0.05 $_{\pm0.02}$ | 0.06 $_{\pm0.03}$ | 0.05 $_{\pm0.03}$ | 0.10 $_{\pm0.02}$ | **0.04** $_{\pm0.02}$ | **0.04** $_{\pm0.02}$ | 0.05 $_{\pm0.02}$ | 0.05 $_{\pm0.03}$ |
| bike | 110.02 $_{\pm1.58}$ | 108.51 $_{\pm1.76}$ | 103.82 $_{\pm2.72}$ | **99.55** $_{\pm1.18}$ | 100.25 $_{\pm1.17}$ | 102.68 $_{\pm1.42}$ | 100.48 $_{\pm1.25}$ | 100.48 $_{\pm1.16}$ | 104.19 $_{\pm1.29}$ | 100.31 $_{\pm1.19}$ |
| house-sales | 0.22 $_{\pm0.01}$ | 0.20 $_{\pm0.00}$ | 0.19 $_{\pm0.00}$ | 0.20 $_{\pm0.00}$ | 0.18 $_{\pm0.00}$ | 0.20 $_{\pm0.01}$ | 0.18 $_{\pm0.00}$ | **0.17** $_{\pm0.00}$ | 0.20 $_{\pm0.00}$ | 0.20 $_{\pm0.00}$ |
| sulfur $_{(10^{-2})}$ | 1.83 $_{\pm0.27}$ | **1.59** $_{\pm0.31}$ | 2.59 $_{\pm0.38}$ | 1.82 $_{\pm0.28}$ | 1.69 $_{\pm0.41}$ | 1.82 $_{\pm0.47}$ | 1.71 $_{\pm0.44}$ | 1.81 $_{\pm0.43}$ | 2.56 $_{\pm0.42}$ | 2.44 $_{\pm0.46}$ |
| Miami2016 | 0.18 $_{\pm0.00}$ | 0.17 $_{\pm0.00}$ | 0.17 $_{\pm0.00}$ | 0.18 $_{\pm0.00}$ | **0.15** $_{\pm0.00}$ | 0.17 $_{\pm0.01}$ | **0.15** $_{\pm0.00}$ | **0.15** $_{\pm0.00}$ | 0.20 $_{\pm0.00}$ | 0.18 $_{\pm0.00}$ |
| superconduct | 11.93 $_{\pm0.26}$ | 9.50 $_{\pm0.22}$ | 14.57 $_{\pm0.44}$ | 11.88 $_{\pm0.26}$ | 9.59 $_{\pm0.20}$ | 11.28 $_{\pm0.90}$ | 13.21 $_{\pm7.04}$ | 10.32 $_{\pm0.22}$ | 13.20 $_{\pm0.17}$ | 10.94 $_{\pm1.08}$ |
| california | 0.16 $_{\pm0.00}$ | 0.16 $_{\pm0.00}$ | 0.16 $_{\pm0.00}$ | 0.15 $_{\pm0.00}$ | **0.12** $_{\pm0.00}$ | 0.15 $_{\pm0.00}$ | 0.13 $_{\pm0.00}$ | 0.13 $_{\pm0.00}$ | 0.17 $_{\pm0.00}$ | 0.14 $_{\pm0.00}$ |
| fifa | 0.84 $_{\pm0.01}$ | 0.84 $_{\pm0.01}$ | 0.79 $_{\pm0.01}$ | 0.82 $_{\pm0.01}$ | 0.82 $_{\pm0.01}$ | 0.81 $_{\pm0.01}$ | 0.81 $_{\pm0.01}$ | 0.80 $_{\pm0.01}$ | **0.77** $_{\pm0.01}$ | 0.78 $_{\pm0.01}$ |

Directly optimising $M$ through MLE in the ARD-Laplace-full is challenging due to the increased complexity associated with often high-dimensional feature spaces. Hence, while the parameterization of both methods is equal, the RFM-based learning method of alternately solving convex problems seems to be simpler to optimise.

## 5.2 TOY DATA SET

Given the qualitatively similar performance of the GP-RFM-Laplace and its diagonal version, we investigate the differences between the two methods in more detail. Mathem-

atically, in the RFM-Laplace-diag we restrict the feature matrix $M$ to be diagonal. Therefore, the RFM-Laplace-diag is a special case of the RFM-Laplace where the latter can additionally capture covariate correlations that are relevant for the predictive task.

To highlight the advantage of the RFM-Laplace, we create a toy dataset. The covariates $x$ are independent and the labels $y$ are nonlinearly transformed using the first 10 covariates

$$x \sim \mathcal{U}(0_d, 1_d); \quad y = (\sum_{j=1}^{10} x_{[j]})^2. \tag{11}$$

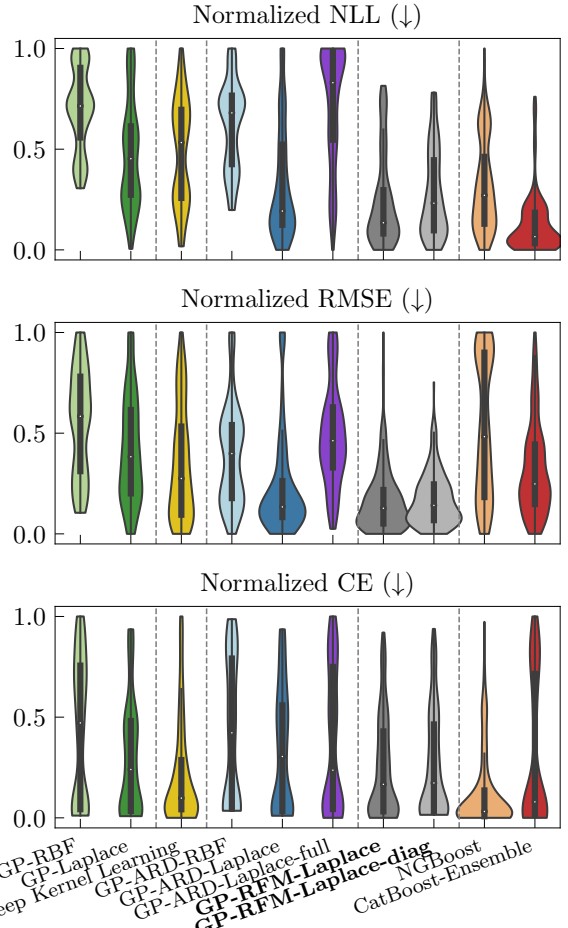

Figure 1: Violin plot results on the OpenML benchmark including boxplots with median and quartiles for each method.

This dataset is crafted to challenge methods that struggle to determine the direction in which covariates are combined, i.e. off-diagonal correlation of covariates. We compare the performance in terms of NLL for a range of feature sizes in Figure 2. The results for the performance in terms of RMSE on all methods can be found in Figure 5.

We observe that the GP-RFM-Laplace outperforms all methods for all covariate dimensions. This demonstrates that a non-diagonal metric in the RFM-Laplace in contrast to diagonal metrics used in kernels with ARD can benefit the performance considerably and has been underexplored in the community. Furthermore, the results in Table 1 and Figure 1 show that no GP-based method outperforms the GP-RFM-Laplace. However, the diagonal kernel with ARD (GP-ARD-Laplace) performs similarly well to our GP-RFM-Laplace for many datasets. Therefore, we conjecture that in many real-world datasets, there is either little covariate correlation or the covariate correlation is not relevant for the predictive task. For datasets where the GP-RFM-Laplace considerably outperforms the GP-ARD-Laplace, such as

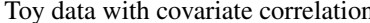
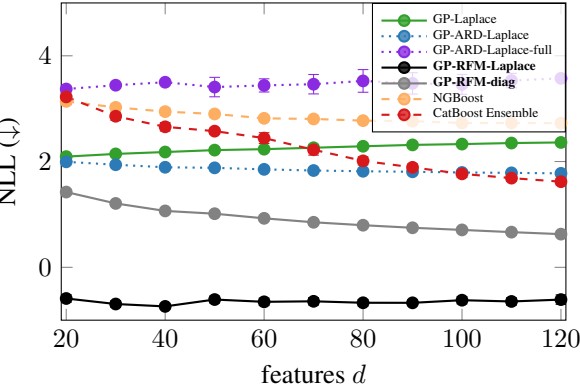

Figure 2: Toy dataset with covariate correlation for prediction. We scale the number of train samples with $n = 20d$.

the 'isolet' (Isolated Letter Speech Recognition) dataset from OpenML, we observe that there is indeed considerable covariate correlation.

## 5.3 VISUALIZING FEATURE MATRICES

To get a better understanding of the learnt feature matrix $M$, we visualize the normalized feature matrices for the RFM-Laplace and its diagonal version RFM-Laplace-diag in Figure 3. On the top row, we compare both methods for the toy dataset, where we generated the labels with correlating covariates according to (11). For this dataset, we can compute the true feature matrix through the Jacobian of the labels with respect to the covariates to obtain the true feature matrix $M$. The true feature matrix is a block matrix with a $10 \times 10$ block of $\frac{1}{n} \sum_{i=1}^{n} (\sum_{j=1}^{10} x_{i[j]})^2$ and the remaining entries are zero, where $x_{i[j]}$ denotes the $j$th dimension of the $i$th sample. It is necessary to learn this non-zero block to capture the relevant covariate correlation. Experimentally, as we expected, in Figure 3 the RFM-Laplace learns relevant covariate correlation as indicated by nonzero off-diagonal values of the feature matrix while the diagonal methods are unable to capture this relation.

On the bottom row, we compare both methods on the Kin8nm dataset from the UCI benchmark. In this real-world dataset, the RFM-Laplace captures the non-zero covariate correlation and focuses on a low-dimensional set of covariates. This ability of RFMs to learn low-dimensional features has been proven for linear RFMs in Radhakrishnan et al. [2024b]. Additionally, we can qualitatively see that the RFM-Laplace learns the same diagonal covariate re-weighing as the RFM-Laplace-diag. Therefore, the RFM-Laplace is a direct generalisation of the RFM-Laplace-diag and can learn more complex features, which allows for both of these datasets to be predicted more accurately.

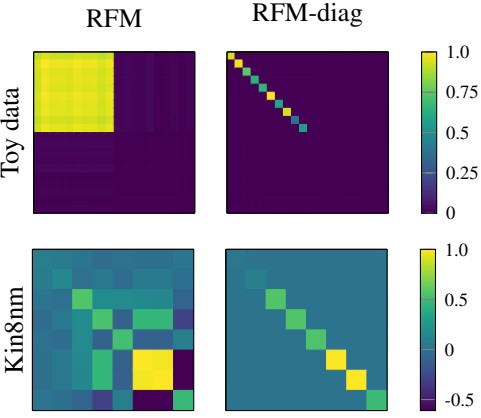

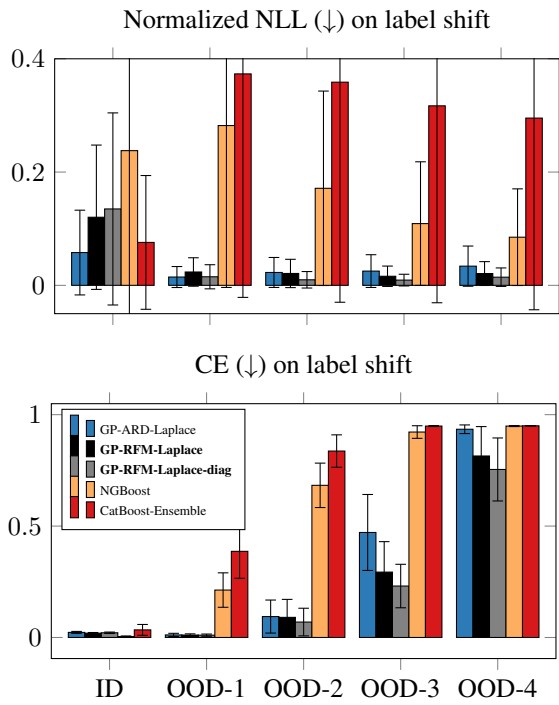

Figure 3: Normalized feature matrices for toy data (top) and Kin8nm dataset from UCI benchmark (bottom).

## 5.4 OUT-OF-DISTRIBUTION DATA

Having established that the GP-RFM-Laplace is a competitive method for probabilistic regression, we now investigate its performance on out-of-distribution (OOD) data. Distribution shift depicts a common scenario in real-world applications where for example the test data distribution changes over time. One hope of utilising a probabilistic model is to obtain more reliable predictions by indicating when the model is uncertain about its predictions. Understanding how well the GP-RFM-Laplace performs in such scenarios is essential for assessing its robustness and applicability in real-world settings.

In our setting, we concentrate on real-world data shifts. Here, we focus on label shifts, i.e. the marginal distribution of the labels $p(y)$ change, while in the Appendix B.7 we also consider covariate shifts, i.e. the marginal distribution of the covariates $p(\boldsymbol{x})$ change. Specifically, we take four house datasets from the OpenML benchmark for which the labels describe the house value and include a covariate for latitude and longitude. We define the ID data such that $p(y > a) = 0.7$ where $a$ is the 70% quantile of the labels and the OOD data such that $p(y < a)$. We then split the OOD data into four consecutive non-overlapping datasets, where each OOD dataset contains 7.5% of the data. This results in one ID dataset and four OOD datasets (denoted with OOD-1 to OOD-4) with increasingly severe label shifts.

Figure 4 shows the results on ID and OOD data for different methods. We notice in the top figure that the NLL of the boosting-based method rises with increasing severity of label shift while the GP-based methods improve. Overall, the GP-based methods including the GP-RFM are the most robust. This reliability is confirmed by the lower CE of the GP-based methods which shows that the model is better calibrated under label shift, see Figure 4 (bottom). We have to note that for large distribution shifts, none of the methods

Figure 4: Out-of-distribution experiment: NLL (top) and CE (bottom) on four house datasets with label shift. We show mean and standard deviation.

are calibrated anymore. Generally, our results indicate that Boosting-based methods are less robust to label shifts as defined in our scenario.

## 6 DISCUSSION AND FUTURE WORK

In this study, we adopted the RFM—a novel data-adaptive, feature learning kernel—for uncertainty quantification through integration into GPs. We rigorously tested our method across various datasets and metrics to ensure consistency. Our results demonstrate that our RFM-based GP can either outperform or match the performance of existing state-of-the-art methods, including boosting-based approaches such as NGBoost [Duan et al., 2020] and CatBoost-ensembles [Malinin et al., 2021].

In the GP literature, there is a focus on ARD-based approaches or low-rank feature matrices $\boldsymbol{M}$ [Garnett et al., 2014, Letham et al., 2020]. We show and provide examples illustrating that the presented GP-RFM with full feature matrix $\boldsymbol{M}$ outperforms these approaches since it is able to reliably model relevant covariate correlation. We therefore bridge fields and demonstrate an approach that the GP community has been missing.

However, our empirical findings suggest that RFMs might occasionally be surpassed by their diagonal version, RFM-diag or kernels with ARD. We observed that sample com-

plexity plays a pivotal role in this behaviour. Given sufficient training samples, leveraging the capabilities of RFM is always preferable. However, in cases where the sample size is limited, the diagonal RFM can be preferable. While delving deeper into determining the optimal method for various settings is beyond the scope of this paper, it presents a crucial direction for future research.

Another line of future research is to integrate more intriguing kernels within the RFM framework. RFM is a broad feature learning framework based on kernels, suitable for any radial kernel. This study primarily concentrates on the two most prevalent kernels: RBF and Laplacian. The results clearly show that the Laplacian outperforms the Gaussian kernel. There is potential to select a task-specific kernel to further enhance these performances. For example, Neural Tangent Kernels (NTK) [Jacot et al., 2018] or Convolutional Neural Tangent Kernels (CNTK) [Li et al., 2019].

Another crucial aspect is scalability. Decision tree-boosting methods are naturally adept at handling large datasets. On the other hand, kernel methods historically have faced challenges in scaling. However, with the advent of recent state-of-the-art techniques, scaling kernels has become feasible. Notable examples are the EigenPro series [Ma and Belkin, 2017, 2019, Abedsoltan et al., 2023, 2024] and FALKON [Rudi et al., 2017, Meanti et al., 2020]. These advancements can enable our method to scale effectively to large datasets.

## Acknowledgements

This work was partially supported by the Wallenberg AI, Autonomous Systems and Software Program (WASP) funded by the Knut and Alice Wallenberg Foundation. The computations were enabled by the Berzelius resource provided by the Knut and Alice Wallenberg Foundation at the National Supercomputer Centre, Sweden.

A.A and M.B are grateful for the support from the National Science Foundation (NSF) and the Simons Foundation for the Collaboration on the Theoretical Foundations of Deep Learning (https://deepfoundations.ai/) through awards DMS-2031883 and #814639 and the TILOS institute (NSF CCF-2112665).

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

# Uncertainty Estimation with Recursive Feature Machines (Supplementary Material)

**Daniel Gedon**[1]  **Amirhesam Abedsoltan**[*2]  **Thomas B. Schön**[1]  **Mikhail Belkin**[2,3]

[1]Department of Information Technology, Uppsala University, Sweden
[2]Department of Computer Science and Engineering, UC San Diego, USA
[3]Halıcıoğlu Data Science Institute, UC San Diego, USA

## CONTENTS

---

[*]Equal contribution.

# A    IMPLEMENTATION DETAILS

## A.1    MODEL AND TRAINING DETAILS

**Gaussian process.**    For the mean, we use a constant function. For the covariance, we use the concatenation of a scale kernel with the respective RBF/Laplace or Mahalanobis distance kernel. All GP-based methods are optimized with Adam optimizer over 250 epochs [Kingma and Ba, 2014] with a cosine annealing learning rate schedule [Loshchilov and Hutter, 2016] to a minimum learning rate of $\text{lr}_{min} = 10^{-7}$ and with all data points in the training set as a mini-batch using GPyTorch [Gardner et al., 2018].

**Deep kernel learning.**    For the mean, we use a constant function. For the covariance, we follow the GPyTorch tutorial implementation of deep kernel learning[1]. Our model consists of a ReLU fully-connected deep neural network with dimensions $\{d, 1000, 500, 50, 2\}$ as in Wilson et al. [2016]. Notably by following the GPyTorch implementation, we do not pre-train the deep neural network. Further, for a fair comparison with other GP-based methods, we did not use the ideas of KISS-GP [Wilson and Nickisch, 2015]. Hence, we use the same optimization scheme as for all other GPs, but we reduced the number of epochs to 100 due to stability issues for longer training schemes.

**GP-ARD-Laplace-full.**    For the mean, we use a constant function. For the covariance, we use the Mahalanobis distance kernel from (7). To stabilize training, we decompose the Mahalanobis distance as $M = U + U^\top + D$. Here, $U$ is a learnable upper triangular matrix of small values to enforce symmetry and the learnable $D$ is a diagonal matrix to focus initialization on the diagonal, similar to the RFM. We use the same optimization scheme as for all other GPs, but we reduced the number of epochs to 150 due to stability issues for longer training schemes.

**NGBoost.**    We use the NGBoost regressor from the official implementations of the respective authors[2]. For this model, we use the default set of hyperparameters.

**CatBoost-ensemble.**    We use the CatBoost regressor from the official implementation of the respective authors[3]. We choose to select 10 ensemble members each consisting of 1000 trees as done similarly in Malinin et al. [2021]. Furthermore, we consider the depth of the trees to be 6 and keep the remaining default hyperparameters

## A.2    HYPERPARAMETER SEARCH

For our main results, we perform a hyperparameter search for the specific hyperparameters of each method. We run a grid search over the hyperparameters and select the best ones based on the validation set's NLL value.

**Gaussian processes.**    The only hyperparameter is the learning rate which we optimize over the values $\text{lr} = \{0.05, 0.01, 0.005, 0.001\}$. The learning rate is decreased to a minimal value of $10^{-7}$.

**Recursive feature machines.**    For the RFM, we optimize the hyperparameters of the GP-based methods as described above. Additionally, we optimize the Ridge regularization for the optimization of $\alpha$ over the values $\lambda_\alpha = \{0.5, 0.1, 0.5, 0.01, 0.001, 0.0001\}$. Furthermore, we optimize the Ridge regularization for the optimization of $M$ over the values $\lambda_M = \{0.1, 0.01, 0.001, 5 \cdot 10^{-5}, 10^{-6}, 10^{-7}, 0\}$.

**Boosting-based.**    For NGBoost, we optimize the number of estimators from $\{100, 200, 300, 400, 500\}$. For CatBoost-ensembles, we optimize the learning rate over the values $\text{lr} = \{0.05, 0.01, 0.005, 0.001\}$

## A.3    POST-PROCESSING

Some methods did not converge for some seeds and datasets. The metrics computed from these runs would heavily distort the actual results but are easy to detect in practice. To evaluate only successful runs, we remove outliers for each dataset and

---

[1] https://docs.gpytorch.ai/en/stable/examples/06_PyTorch_NN_Integration_DKL/KISSGP_Deep_Kernel_Regression_CUDA.html, accessed 05.02.2024.
[2] https://stanfordmlgroup.github.io/ngboost/, accessed 05.02.2024.
[3] https://catboost.ai/, accessed 05.02.2024.

metric individually. We achieve this by removing entries with z-values $\geq 3.5$. Almost exclusively the results from deep kernel learning and the GP-ARD-Laplace-full are affected by this post-processing.

# B  ADDITIONAL EXPERIMENTAL RESULTS

## B.1  COMPUTATIONAL INFRASTRUCTURE

All experiments are run on a single NVIDIA A100 GPU with 40GB memory. The GPU is part of an internal cluster supported by local resources. To run the experiments for all methods on all datasets in a sequence of our used OpenML benchmark, we require approximately 1 hour of computation time for one seed. For all methods on all datasets in a sequence of the UCI benchmark, we require approximately 10 minutes of computation time for one seed.

## B.2  MAIN RESULTS

Complementary to the main results, in Figure 6 we list the normalized results for the OpenML benchmark and the UCI benchmark. Qualitatively the observations from the OpenML benchmark also hold for the UCI benchmark.

Additionally to the metrics NLL, RMSE and CE, in Figures 6c and 6d we show the combined time for training the model and prediction on the test set. We have to note that the GP-based methods utilize the GPytorch library which enables GPU utilization. For NGBoost and CatBoost-ensembles, the official implementations do not allow for GPU utilization. Therefore, the time comparison is on the one side biased because we utilize different hardware, on the other side it utilizes the best openly available implementations for all respective methods. Note, that here, we excluded the method 'deep Kernel learning'. This method reduces the GP dimensionality to 2 dimensions because of the neural networks extractor and is therefore the fastest method. Including it in the violin plot would distort the visualization. Detailed timing results for each method on every dataset can be found in Appendix C.

## B.3  TOY DATA SET

We show results from the toy dataset which was designed to be difficult for methods which do not capture feature correlation. This includes diagonal feature weighting such as the ARD-based methods. We included correlation to be modelled into the data by choosing $\boldsymbol{x} = \mathcal{U}(0_d, 1_d)$ and $y = (\sum_{i=1}^{1} 0\boldsymbol{x}_{[i]})^2$. In Figure 5, we plot additionally to the NLL also the RMSE which show a qualitatively similar behaviour such that the GP-RFM-Laplace outperforms other methods. Notably the diagonal method, GP-RFM-diag becomes close for high feature dimensions. We argue that this is because we fix the number of dimensions which are relevant for the prediction but grow the actual dimension. Hence, fewer relative dimensions become relevant. However, the diagonal method will never outperform the GP-RFM-Laplace on this toy dataset.

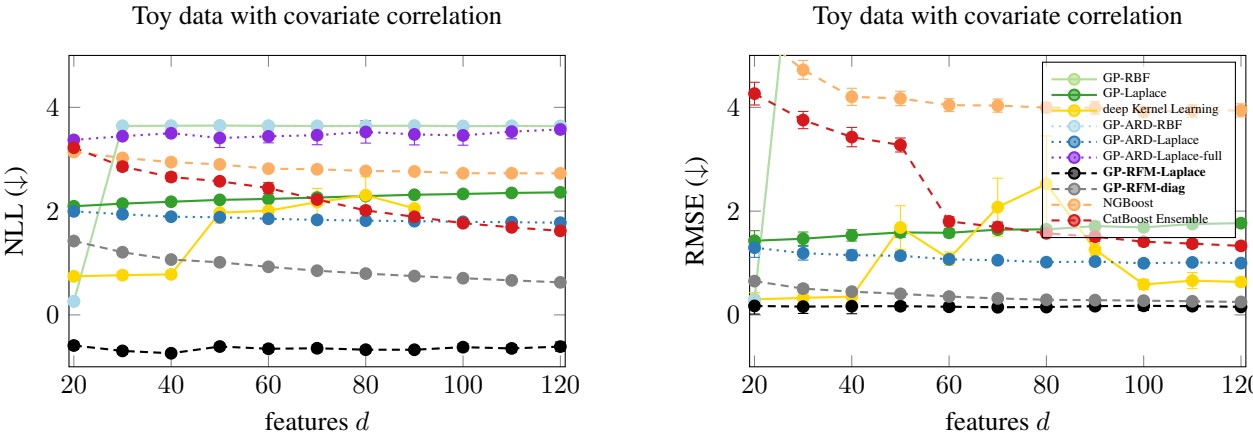

Figure 5: Toy data set with varying feature dimensions. NLL (left) is a repetition of the main text figure; RMSE (right) shows a similar pattern. Note that for NLL the deep Kernel Learning blows up at $d = 100$, hence these values are omitted. Similarly for RMSE, the GP-ARD-Laplace-full $> 5$ and therefore not depicted.

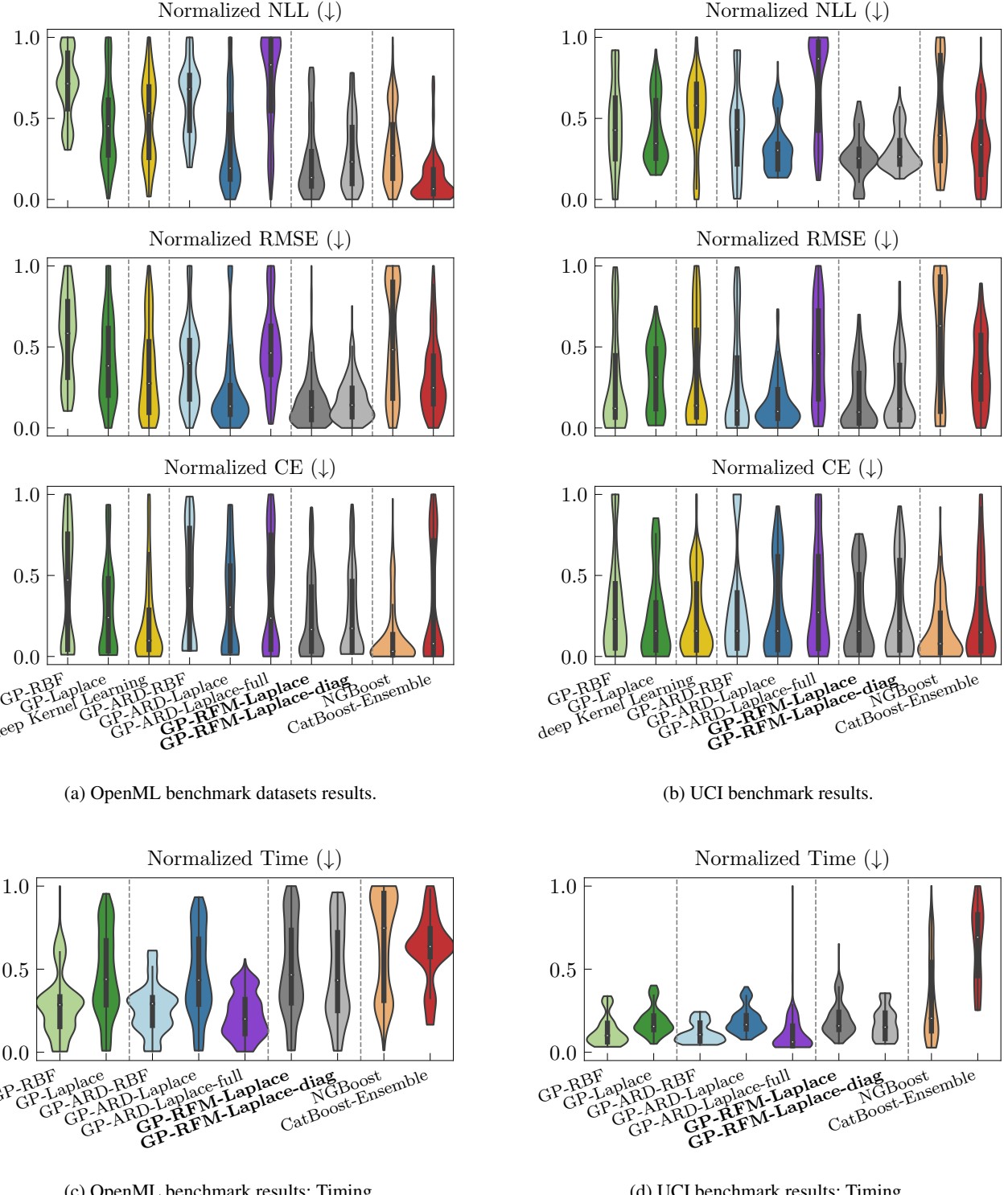

(a) OpenML benchmark datasets results.

(b) UCI benchmark results.

(c) OpenML benchmark results: Timing.

(d) UCI benchmark results: Timing.

Figure 6: Violin plot results on OpenML benchmark datasets and UCI benchmark. Note that Figure 6a is a repetition of Figure 1 from the main text. Note also that in Figures 6c and 6d we excluded the method 'deep Kernel Learning' since it is the fastest and distorts the visualization in the violin plots.

## B.4  VISUALIZING FEATURE MATRICES

In Figure 7, we compare the learnt feature matrices $M$ of the RFM-based methods with the ones from Kernels with ARD on the toy dataset. The only method which can capture the necessary feature correlation is the GP-RFM-Laplace. Notably, the GP-ARD-Laplace-full is not able to learn the correlation despite its structure ability through parameterization with a full feature matrix $M$. This might be justified by the more complicated optimization problem resulting in poor performance as Figure 5 indicates for this method.

For the diagonal methods, the GP-RFM-Laplace-diag capture the exact dimensionality of the problem by weighting all irrelevant dimensions in the toy problem with zeros. In contrast, the GP-ARD-Laplace also capture the necessary dimensions but does not suppress irrelevant dimensions to zero. We experimented with increasing the compute budget for this method from 250 epochs up to 2,000 epochs which reduces the weighting of the irrelevant dimensions but does get close to zero weighting.

Similar conclusions can be drawn about the bottom row for real data. Again, we observe that the GP-ARD-Laplace-full struggles with the task. The remaining three methods learn similar features on the diagonal but only the GP-RFM-Laplace can model the necessary correlation to achieve the best performance.

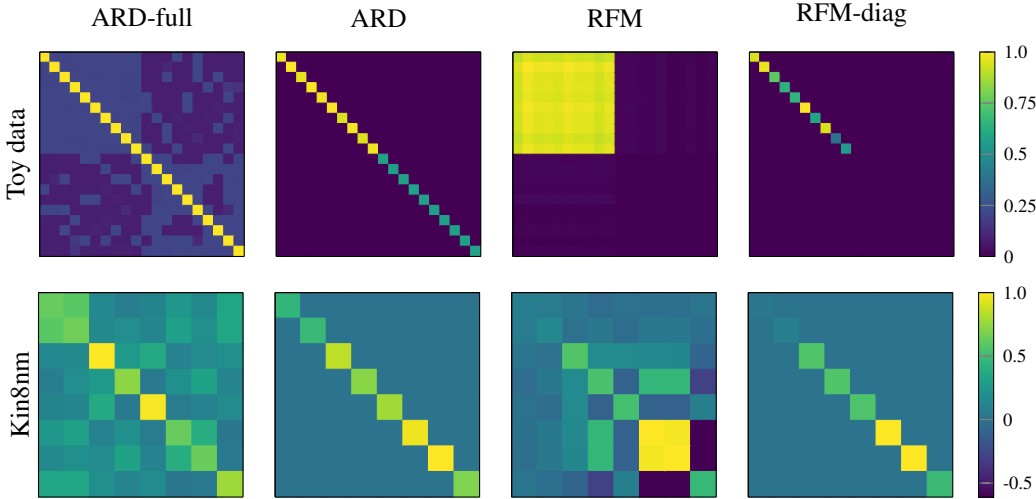

Figure 7: Normalized feature matrices for toy data (top) and Kin8nm dataset from UCI benchmark (bottom). Note that the two right-most columns are a repetition of Figure 3.

## B.5  COMPARISON OF LEARNT FEATURES

Given the comparable performance of the GP-RFM-Laplace and the GP-ARD-Laplace and their similar mathematical structure based on the Mahalanobis distance kernel, a question arises whether the learnt features in $M$ are similar. To investigate this, we compare the Pearson and Spearman correlation between the feature matrices $M$ learnt in RFM-based kernels and kernels with ARD:

- **Diagonal methods:** We compare diagonal of $M$ from the GP-RFM-Laplace-diag with GP-ARD-Laplace.
- **Non-diagonal methods:** Here we perform two comparisons. (1) we compare the full feature matrix $M$ of GP-RFM-Laplace with the one of GP-ARD-Laplace-full. (2) to capture how the features are re-weighted, we also compare the diagonal of the full feature matrix $M$ of both methods.

Comparing diagonal methods separately from methods which learn the full $M$ to disentangle the effects of the parameterization and the learning paradigm. We compare methods with the same parameterization Hence, the main difference lies in the feature learning procedure: the RFM-based kernels learn the features through AGOP iterations while the ARD-based kernels learn the features through MLE optimization.

Figure 8 shows the Pearson and Spearman correlation between the diagonal methods and Figure 9 on the non-diagonal methods on the UCI benchmark dataset. There is the same trend for the diagonal and the non-diagonal methods: For some

datasets, there is a high correlation between the RFM-based kernel and the kernels with ARD, but there are also datasets where the feature correlation is low. This indicates that learning the features with AGOP iterations in the RFM or with MLE optimization in the ARD-based kernel may result in the same features in some cases but is not guaranteed to do so. The similarity between the learning paradigms opens up investigations of the widely applied MLE framework from a different perspective. Further investigation is required to understand the differences between feature learning in the two paradigms.

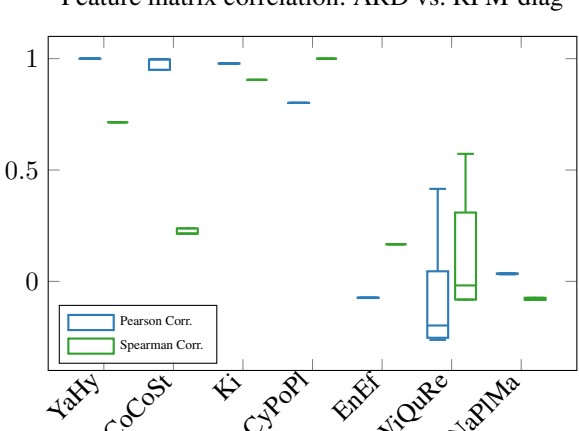

Figure 8: Correlation of feature matrices $M$ between diagonal methods GP-RFM-Laplace-diag and GP-ARD-Laplace.

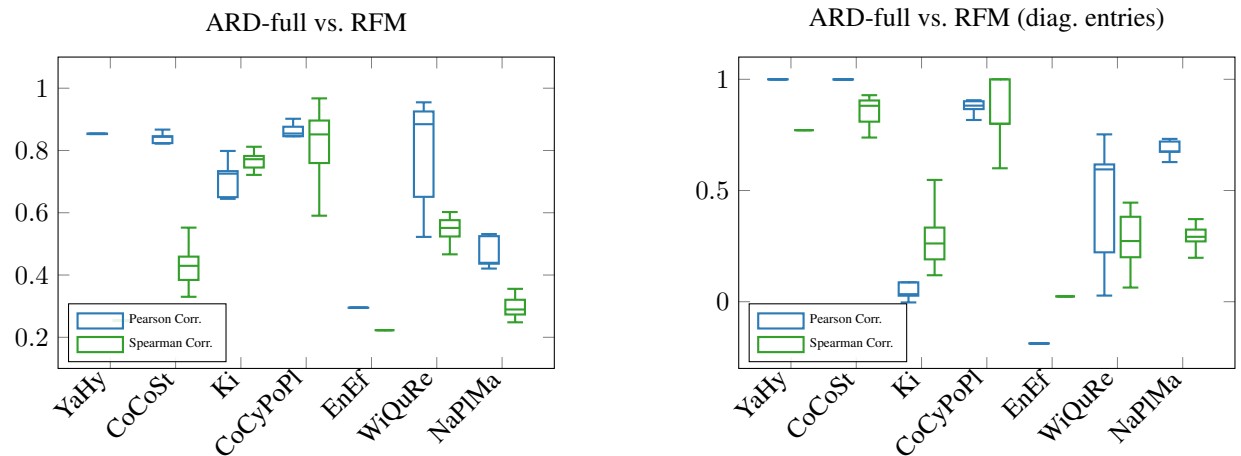

Figure 9: Correlation of feature matrices $M$ between non-diagonal methods GP-RFM-Laplace and GP-ARD-Laplace-full. Left: full $M$ of both methods. Right: diagonal $M$ of both methods.

## B.6 FEATURE IMPORTANCE

Here, we analyse if the features learnt by the RFM are meaningful in the sense of weighting the covariates with the highest predictive power. We consider the feature matrix $M$ of a trained GP-RFM-Laplace and successively remove the covariates with the highest weight of $\text{diag}(M)$. Then, we re-train and evaluate all models on the reduced dataset. Specifically, we use the 'pol' dataset from the OpenML benchmarks since it has a high number of covariates (26). In Figure 10 we observe that the NLL and RMSE increase for all methods when removing the most important covariates but this plateaus. Additionally, we observe that the two most important covariates according to the RFM feature matrix can be removed without a significant increase in NLL and RMSE. This indicates that for this dataset the two highest weighted covariates are equally important for the prediction and contain most of the predictive power. The sharp increase in NLL and RMSE after removing more covariates indicates that the RFM feature matrix can identify the most important covariates for the prediction.

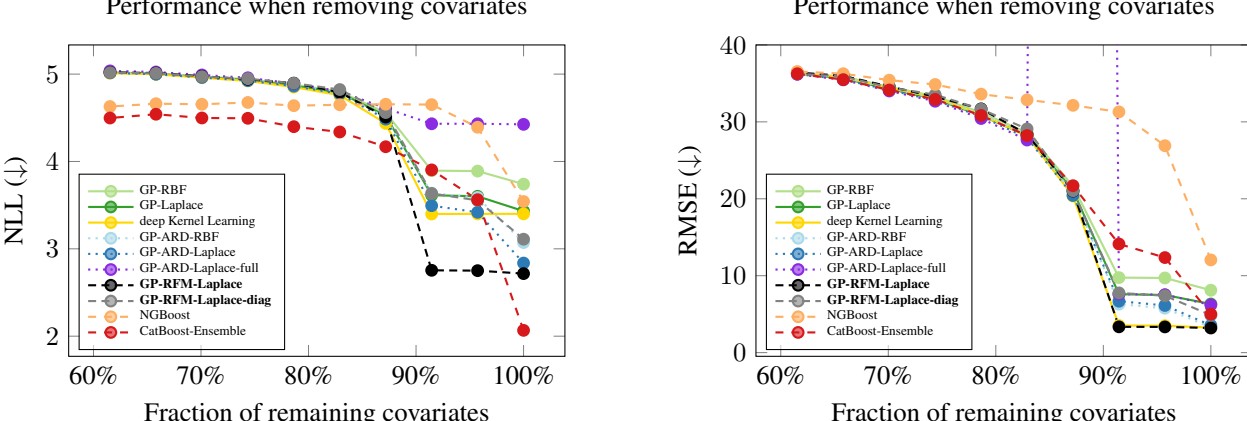

Figure 10: NLL (left) and RMSE (right) when removing the most important covariates according to the diagonal of the RFM feature matrix.

## B.7 DISTRIBUTION SHIFT

In the main text, we show the results of label shift in terms of normalized NLL and CE for some selected methods. Here, we additionally present results for normalized RMSE and normalized interval length in Figure 11. Furthermore, we experimented with covariate shifts. Specifically, we consider the covariate for the latitude of the house location. Similarly to the label shift we define the ID data such that $p(x_{lat} < a) = 0.7$ where $a$ is the $70\%$ quantile of the labels and the OOD data such that $p(x_{lat} > a)$. We split the OOD data into four consecutive non-overlapping datasets, where each contains $7.5\%$ of the data. The results for the covariate shift over the four house datasets are in Figure 12. The results are qualitatively similar to the results on label shift in Figure 11.

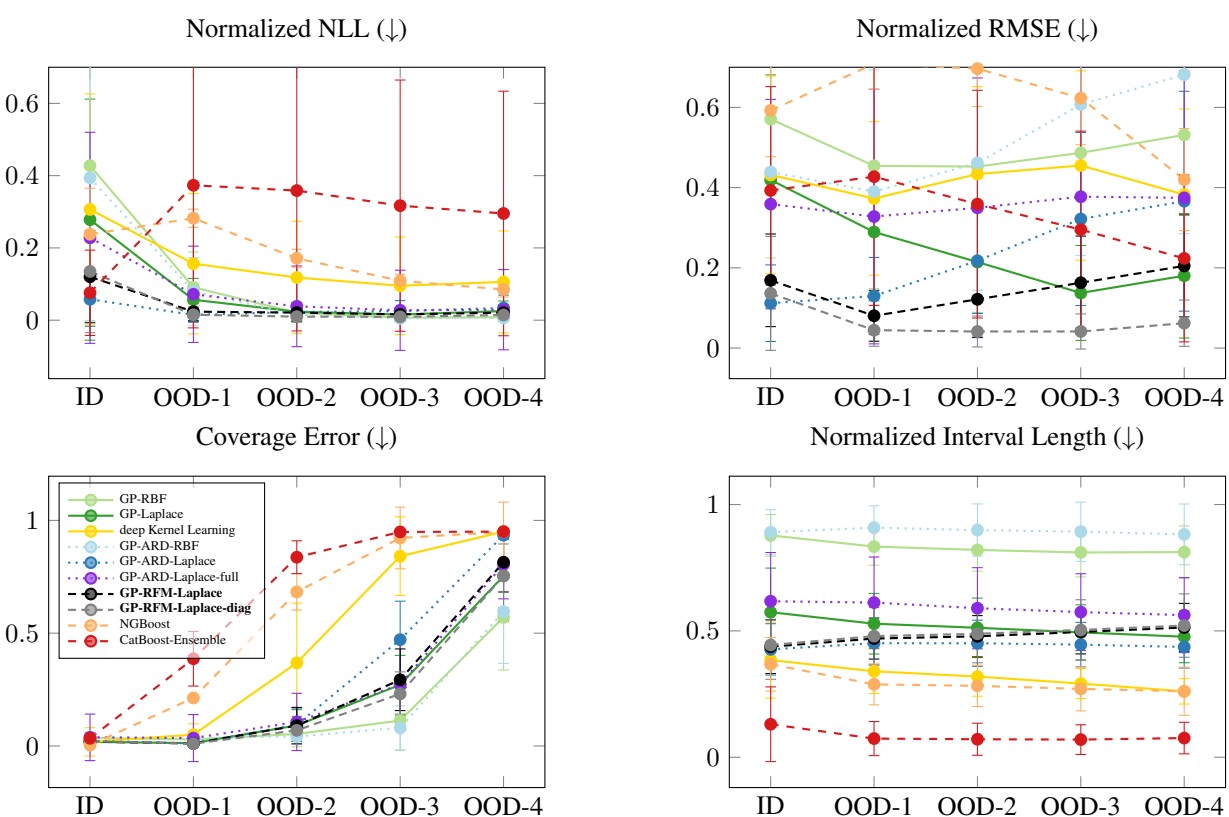

Figure 11: Label shift on four house datasets from the OpenML benchmark.

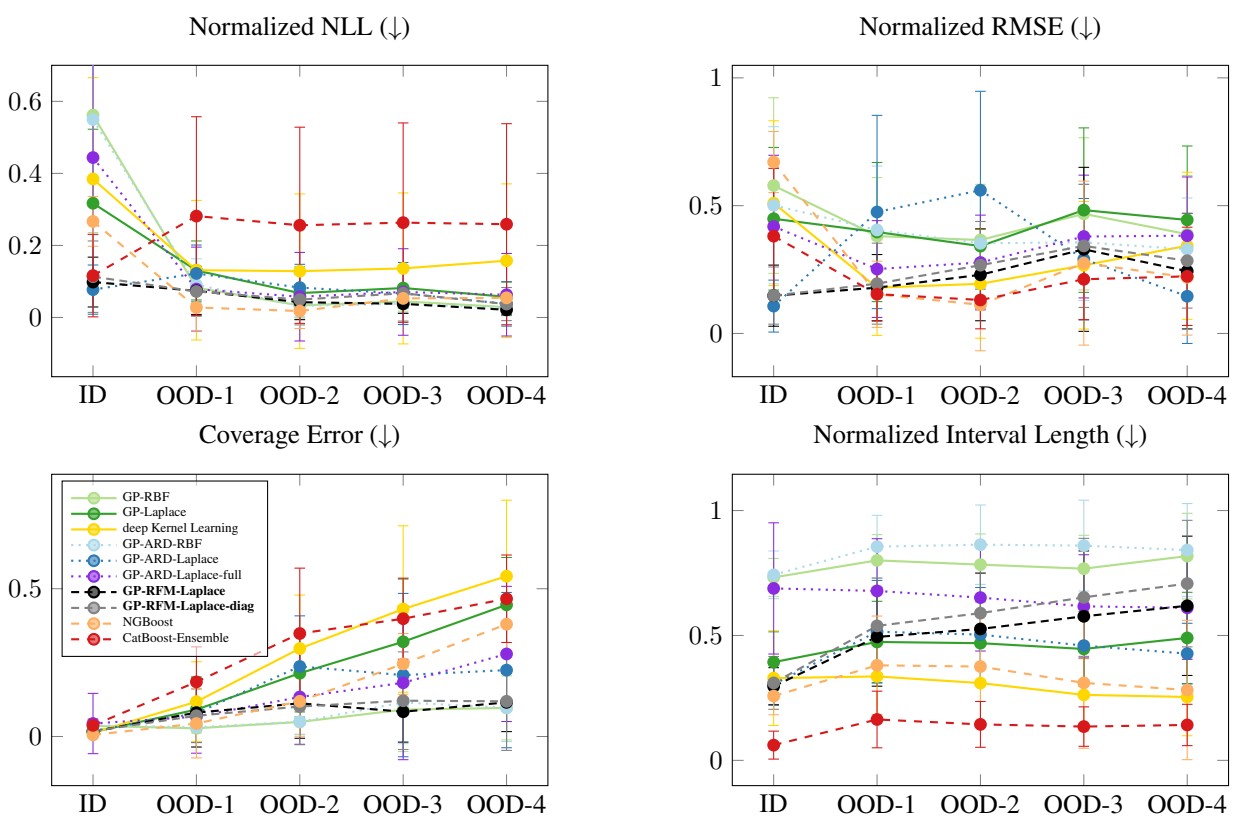

Figure 12: Covariate shift (we use the covariate for the latitude of the house position) on four house datasets from the OpenML benchmark.

# C   DETAILED RESULTS ON TABULAR BENCHMARKS

In the main text and Figure 6 we show summary results over all datasets of the two benchmarks we consider. The following tables present the results individually for each dataset in both benchmarks.

## C.1   TABULAR BENCHMARK

Table 3: Time ($\downarrow$) in seconds required for training and prediction on the OpenML benchmark.

| Dataset | Gaussian Process | | | | | | Ours | | Boosting | |
| | RBF | Laplace | deep KL | ARD RBF | Lap. | Lap.-full | RFM | RFM-diag | NGBoost | CatBoost |
|---|---|---|---|---|---|---|---|---|---|---|
| cpu-act | 13.23 $\pm$0.76 | 10.84 $\pm$0.59 | 2.87 $\pm$0.06 | 10.59 $\pm$0.30 | 10.77 $\pm$0.56 | 8.88 $\pm$0.30 | 10.92 $\pm$0.36 | 10.31 $\pm$0.24 | 23.03 $\pm$3.33 | 27.37 $\pm$4.10 |
| pol | 12.47 $\pm$0.17 | 17.30 $\pm$0.50 | 4.49 $\pm$0.05 | 13.77 $\pm$0.25 | 17.08 $\pm$0.45 | 10.56 $\pm$0.68 | 17.84 $\pm$0.50 | 17.54 $\pm$0.08 | 18.92 $\pm$0.17 | 24.16 $\pm$3.76 |
| elevators | 22.38 $\pm$0.53 | 20.37 $\pm$0.20 | 5.24 $\pm$0.05 | 20.79 $\pm$0.19 | 20.39 $\pm$0.24 | 12.35 $\pm$0.12 | 21.73 $\pm$0.32 | 21.45 $\pm$0.03 | 23.83 $\pm$0.15 | 23.50 $\pm$2.59 |
| isolet | 6.69 $\pm$0.08 | 11.42 $\pm$0.73 | 3.04 $\pm$0.05 | 7.85 $\pm$0.08 | 11.86 $\pm$0.49 | 6.47 $\pm$0.15 | 12.84 $\pm$0.75 | 10.79 $\pm$0.16 | 830.81 $\pm$3.03 | 258.70 $\pm$17.49 |
| wine | 6.80 $\pm$0.15 | 11.15 $\pm$0.56 | 2.67 $\pm$0.03 | 6.86 $\pm$0.17 | 10.95 $\pm$1.32 | 5.69 $\pm$0.22 | 11.23 $\pm$0.60 | 9.43 $\pm$0.80 | 9.97 $\pm$1.11 | 21.83 $\pm$3.37 |
| Ailerons | 14.13 $\pm$0.26 | 15.66 $\pm$1.03 | 4.22 $\pm$0.06 | 14.05 $\pm$0.15 | 15.46 $\pm$0.67 | 9.45 $\pm$0.41 | 16.45 $\pm$1.11 | 14.96 $\pm$0.23 | 29.33 $\pm$0.17 | 27.14 $\pm$2.65 |
| houses | 15.65 $\pm$0.24 | 32.46 $\pm$1.29 | 6.53 $\pm$0.06 | 15.75 $\pm$0.19 | 32.30 $\pm$0.65 | 19.35 $\pm$0.38 | 34.40 $\pm$1.05 | 33.79 $\pm$0.23 | 31.97 $\pm$0.20 | 27.83 $\pm$2.63 |
| houses-16H | 20.87 $\pm$6.45 | 39.30 $\pm$0.62 | 7.48 $\pm$0.30 | 22.29 $\pm$7.01 | 39.13 $\pm$0.13 | 23.73 $\pm$0.41 | 41.80 $\pm$0.98 | 41.43 $\pm$0.06 | 72.42 $\pm$5.38 | 32.03 $\pm$3.35 |
| Bra-houses | 17.35 $\pm$6.80 | 12.31 $\pm$0.57 | 3.27 $\pm$0.04 | 9.72 $\pm$0.72 | 12.48 $\pm$1.06 | 6.86 $\pm$0.32 | 12.80 $\pm$0.82 | 11.57 $\pm$0.55 | 11.51 $\pm$0.52 | 23.42 $\pm$2.40 |
| bike | 12.42 $\pm$0.08 | 22.44 $\pm$0.14 | 5.56 $\pm$0.02 | 12.80 $\pm$0.07 | 22.51 $\pm$0.21 | 13.39 $\pm$0.19 | 24.37 $\pm$1.67 | 23.53 $\pm$0.07 | 12.81 $\pm$0.16 | 20.11 $\pm$2.73 |
| house-sales | 24.78 $\pm$3.06 | 35.02 $\pm$0.16 | 6.84 $\pm$0.05 | 22.93 $\pm$0.40 | 35.05 $\pm$0.23 | 21.27 $\pm$0.53 | 37.50 $\pm$0.52 | 37.13 $\pm$0.08 | 39.21 $\pm$0.24 | 29.78 $\pm$3.38 |
| sulfur | 7.69 $\pm$0.07 | 12.25 $\pm$1.49 | 3.38 $\pm$0.12 | 7.84 $\pm$0.09 | 12.62 $\pm$1.49 | 6.18 $\pm$0.19 | 12.99 $\pm$1.21 | 10.85 $\pm$0.33 | 11.36 $\pm$4.35 | 25.19 $\pm$4.00 |
| Miami2016 | 10.26 $\pm$1.44 | 15.49 $\pm$0.54 | 4.37 $\pm$0.05 | 10.18 $\pm$0.06 | 15.46 $\pm$0.64 | 8.97 $\pm$0.33 | 16.65 $\pm$0.79 | 15.32 $\pm$0.25 | 38.62 $\pm$0.23 | 30.22 $\pm$4.17 |
| superconduct | 15.88 $\pm$0.08 | 34.60 $\pm$0.66 | 6.78 $\pm$0.04 | 17.04 $\pm$0.07 | 34.47 $\pm$0.58 | 28.85 $\pm$33.30 | 71.69 $\pm$83.62 | 37.35 $\pm$0.98 | 262.93 $\pm$1.43 | 61.23 $\pm$5.36 |
| california | 16.38 $\pm$0.25 | 32.44 $\pm$0.82 | 6.49 $\pm$0.13 | 16.13 $\pm$0.25 | 32.34 $\pm$0.93 | 19.49 $\pm$0.11 | 34.37 $\pm$0.77 | 33.98 $\pm$0.48 | 35.96 $\pm$0.26 | 28.92 $\pm$3.32 |
| fifa | 13.99 $\pm$0.62 | 24.25 $\pm$0.12 | 5.58 $\pm$0.04 | 14.59 $\pm$0.15 | 24.40 $\pm$0.30 | 14.50 $\pm$0.08 | 26.26 $\pm$1.38 | 25.72 $\pm$0.07 | 14.29 $\pm$0.65 | 25.10 $\pm$3.56 |

Table 4: OpenML dataset: cpu-act (8192 samples; 21 covariates)

| | RMSE ($\downarrow$) | NLL ($\downarrow$) | CE (95%) ($\downarrow$) | IL (95%) ($\downarrow$) | Time ($\downarrow$) |
|---|---|---|---|---|---|
| GP-RBF | 4.1983 $\pm$0.3682 | 2.8004 $\pm$0.0687 | 0.0407 $\pm$0.0030 | 29.0699 $\pm$2.8943 | 13.2314 $\pm$0.7587 |
| GP-Laplace | 3.8907 $\pm$0.2875 | 2.5481 $\pm$0.0244 | 0.0310 $\pm$0.0029 | 15.5996 $\pm$0.1433 | 10.8411 $\pm$0.5874 |
| deep Kernel Learning | 2.5007 $\pm$0.0820 | 2.6724 $\pm$0.0315 | 0.0892 $\pm$0.1866 | 20.1186 $\pm$0.7163 | 2.8708 $\pm$0.0596 |
| GP-ARD-RBF | 3.3840 $\pm$0.2889 | 2.6651 $\pm$0.0397 | 0.0430 $\pm$0.0019 | 23.5181 $\pm$1.1309 | 10.5879 $\pm$0.0596 |
| GP-ARD-Laplace | 2.7398 $\pm$0.1460 | 2.2968 $\pm$0.0173 | 0.0279 $\pm$0.0029 | 11.7734 $\pm$0.0615 | 10.7739 $\pm$0.5581 |
| GP-ARD-Laplace-full | 7.1210 $\pm$1.0766 | 3.7143 $\pm$0.0963 | 0.0376 $\pm$0.0037 | 59.3472 $\pm$3.7178 | 8.8822 $\pm$0.2975 |
| **GP-RFM-Laplace** | 2.3291 $\pm$0.1149 | 2.2050 $\pm$0.0131 | 0.0206 $\pm$0.0032 | 10.0737 $\pm$0.1053 | 10.9167 $\pm$0.3581 |
| **GP-RFM-Laplace-diag** | 2.1579 $\pm$0.0447 | 2.1661 $\pm$0.0091 | 0.0247 $\pm$0.0028 | 9.8702 $\pm$0.0921 | 10.3116 $\pm$0.2368 |
| NGBoost | 2.4774 $\pm$0.0915 | 2.3276 $\pm$0.1373 | 0.0077 $\pm$0.0057 | 8.5031 $\pm$0.7630 | 23.0311 $\pm$3.3259 |
| CatBoost-Ensemble | 2.5011 $\pm$0.1260 | 2.1699 $\pm$0.0295 | 0.0081 $\pm$0.0042 | 8.0882 $\pm$0.0825 | 27.3657 $\pm$4.0987 |

Table 5: OpenML dataset: pol (15000 samples; 26 covariates)

| | RMSE (↓) | NLL (↓) | CE (95%) (↓) | IL (95%) (↓) | Time (↓) |
|---|---|---|---|---|---|
| GP-RBF | 8.1744 ±0.2212 | 3.7279 ±0.0083 | 0.0402 ±0.0012 | 60.3733 ±0.5259 | 12.4684 ±0.1706 |
| GP-Laplace | 6.4119 ±0.0992 | 3.4294 ±0.0064 | 0.0281 ±0.0021 | 40.8013 ±0.2545 | 17.3032 ±0.4992 |
| deep Kernel Learning | 3.3911 ±0.3008 | 3.4039 ±0.0125 | 0.0462 ±0.0017 | 44.8599 ±0.9984 | 4.4946 ±0.0514 |
| GP-ARD-RBF | 3.2847 ±0.0940 | 3.0713 ±0.0065 | 0.0462 ±0.0006 | 33.9202 ±0.3687 | 13.7665 ±0.0514 |
| GP-ARD-Laplace | 3.5131 ±0.1060 | 2.8353 ±0.0070 | 0.0330 ±0.0021 | 22.9351 ±0.1555 | 17.0843 ±0.4549 |
| GP-ARD-Laplace-full | 6.3502 ±0.2260 | 4.4107 ±0.0210 | 0.2220 ±0.3442 | 127.1110 ±1.5457 | 10.5559 ±0.6841 |
| **GP-RFM-Laplace** | 3.3721 ±0.1531 | 2.7280 ±0.0144 | 0.0301 ±0.0024 | 19.6895 ±0.1998 | 17.8368 ±0.5048 |
| **GP-RFM-Laplace-diag** | 4.7450 ±0.1514 | 3.1021 ±0.0125 | 0.0269 ±0.0021 | 29.3767 ±0.2548 | 17.5354 ±0.0763 |
| NGBoost | 12.0208 ±0.2321 | 3.5523 ±0.0134 | 0.0150 ±0.0023 | 42.2595 ±0.4839 | 18.9153 ±0.1735 |
| CatBoost-Ensemble | 4.9089 ±0.1442 | 2.0865 ±0.0324 | 0.0136 ±0.0041 | 10.6416 ±0.1790 | 24.1609 ±3.7644 |

Table 6: OpenML dataset: elevators (16599 samples; 16 covariates)

| | RMSE (↓) | NLL (↓) | CE (95%) (↓) | IL (95%) (↓) | Time (↓) |
|---|---|---|---|---|---|
| GP-RBF | 0.0022 ±0.0000 | -4.4642 ±0.0313 | 0.0400 ±0.0017 | 0.0166 ±0.0008 | 22.3781 ±0.5291 |
| GP-Laplace | 0.0022 ±0.0000 | -4.6729 ±0.0090 | 0.0259 ±0.0023 | 0.0105 ±0.0001 | 20.3693 ±0.1979 |
| deep Kernel Learning | 0.0019 ±0.0000 | -4.8461 ±0.0123 | 0.0109 ±0.0031 | 0.0081 ±0.0002 | 5.2406 ±0.0463 |
| GP-ARD-RBF | 0.0022 ±0.0001 | -4.5271 ±0.0177 | 0.0404 ±0.0014 | 0.0151 ±0.0004 | 20.7872 ±0.0463 |
| GP-ARD-Laplace | 0.0021 ±0.0000 | -4.7496 ±0.0092 | 0.0244 ±0.0030 | 0.0097 ±0.0001 | 20.3881 ±0.2434 |
| GP-ARD-Laplace-full | 0.0026 ±0.0002 | -4.3103 ±0.1588 | 0.0397 ±0.0044 | 0.0181 ±0.0033 | 12.3529 ±0.1221 |
| **GP-RFM-Laplace** | 0.0019 ±0.0000 | -4.8622 ±0.0085 | 0.0154 ±0.0025 | 0.0081 ±0.0001 | 21.7328 ±0.3192 |
| **GP-RFM-Laplace-diag** | 0.0020 ±0.0000 | -4.7850 ±0.0086 | 0.0205 ±0.0024 | 0.0092 ±0.0001 | 21.4464 ±0.0321 |
| NGBoost | 0.0036 ±0.0001 | -4.4798 ±0.0151 | 0.0047 ±0.0030 | 0.0115 ±0.0001 | 23.8299 ±0.1544 |
| CatBoost-Ensemble | 0.0023 ±0.0000 | -4.7321 ±0.0183 | 0.0451 ±0.0042 | 0.0068 ±0.0001 | 23.4951 ±2.5942 |

Table 7: OpenML dataset: isolet (7797 samples; 613 covariates)

| | RMSE (↓) | NLL (↓) | CE (95%) (↓) | IL (95%) (↓) | Time (↓) |
|---|---|---|---|---|---|
| GP-RBF | 7.5039 ±0.0491 | 3.4345 ±0.0065 | 0.0500 ±0.0000 | 29.4934 ±0.1699 | 6.6856 ±0.0760 |
| GP-Laplace | 7.5039 ±0.0491 | 3.4344 ±0.0065 | 0.0500 ±0.0000 | 29.5290 ±0.0978 | 11.4205 ±0.7336 |
| deep Kernel Learning | 3.1464 ±0.2413 | 2.6187 ±0.0857 | 0.0227 ±0.0090 | 11.6625 ±3.1630 | 3.0425 ±0.0533 |
| GP-ARD-RBF | 7.5039 ±0.0491 | 3.4345 ±0.0065 | 0.0500 ±0.0000 | 29.4947 ±0.1697 | 7.8513 ±0.0533 |
| GP-ARD-Laplace | 7.5039 ±0.0490 | 3.4345 ±0.0065 | 0.0500 ±0.0000 | 29.5526 ±0.1406 | 11.8644 ±0.4897 |
| GP-ARD-Laplace-full | 7.5039 ±0.0490 | 3.4345 ±0.0065 | 0.0500 ±0.0000 | 29.5522 ±0.0904 | 6.4689 ±0.1459 |
| **GP-RFM-Laplace** | 2.5696 ±0.0979 | 2.3408 ±0.0373 | 0.0057 ±0.0043 | 9.9181 ±0.3743 | 12.8427 ±0.7516 |
| **GP-RFM-Laplace-diag** | 3.1757 ±0.0768 | 2.5661 ±0.0167 | 0.0102 ±0.0042 | 14.2781 ±0.2057 | 10.7894 ±0.1638 |
| NGBoost | 4.1270 ±0.0689 | 2.7053 ±0.0157 | 0.0041 ±0.0028 | 14.5239 ±0.1299 | 830.8086 ±3.0250 |
| CatBoost-Ensemble | 3.4882 ±0.0693 | 2.5194 ±0.0215 | 0.0395 ±0.0072 | 10.1068 ±0.1052 | 258.7036 ±17.4887 |

Table 8: OpenML dataset: wine-quality (6497 samples; 11 covariates)

| | RMSE (↓) | NLL (↓) | CE (95%) (↓) | IL (95%) (↓) | Time (↓) |
|---|---|---|---|---|---|
| GP-RBF | 0.6684 ±0.0190 | 1.0380 ±0.0237 | 0.0152 ±0.0060 | 3.1303 ±0.1727 | 6.8032 ±0.1487 |
| GP-Laplace | 0.6086 ±0.0172 | 0.9493 ±0.0152 | 0.0176 ±0.0051 | 2.9316 ±0.0346 | 11.1491 ±0.5616 |
| deep Kernel Learning | 0.6958 ±0.0129 | 1.0632 ±0.0235 | 0.0178 ±0.0139 | 2.5838 ±0.1521 | 2.6719 ±0.0273 |
| GP-ARD-RBF | 0.6699 ±0.0208 | 1.0434 ±0.0258 | 0.0172 ±0.0065 | 3.1711 ±0.1556 | 6.8614 ±0.0273 |
| GP-ARD-Laplace | 0.6114 ±0.0175 | 0.9496 ±0.0155 | 0.0165 ±0.0063 | 2.8888 ±0.0613 | 10.9450 ±1.3206 |
| GP-ARD-Laplace-full | 0.6129 ±0.0166 | 0.9488 ±0.0166 | 0.0136 ±0.0061 | 2.8646 ±0.0338 | 5.6883 ±0.2169 |
| **GP-RFM-Laplace** | 0.6105 ±0.0170 | 0.9494 ±0.0151 | 0.0168 ±0.0053 | 2.8991 ±0.0597 | 11.2258 ±0.6049 |
| **GP-RFM-Laplace-diag** | 0.6132 ±0.0161 | 0.9523 ±0.0154 | 0.0170 ±0.0058 | 2.8860 ±0.0737 | 9.4255 ±0.8003 |
| NGBoost | 0.6981 ±0.0153 | 1.0351 ±0.0256 | 0.0135 ±0.0082 | 2.5237 ±0.0325 | 9.9671 ±1.1130 |
| CatBoost-Ensemble | 0.6910 ±0.0159 | 1.0276 ±0.0273 | 0.0201 ±0.0124 | 2.4433 ±0.1155 | 21.8260 ±3.3698 |

Table 9: OpenML dataset: Ailerons (13750 samples; 33 covariates)

| | RMSE (↓) | NLL (↓) | CE (95%) (↓) | IL (95%) (↓) | Time (↓) |
|---|---|---|---|---|---|
| GP-RBF | 0.0002 ±0.0000 | -7.1758 ±0.0275 | 0.0341 ±0.0028 | 0.0010 ±0.0000 | 14.1305 ±0.2561 |
| GP-Laplace | 0.0002 ±0.0000 | -7.3105 ±0.0092 | 0.0122 ±0.0029 | 0.0007 ±0.0000 | 15.6566 ±1.0279 |
| deep Kernel Learning | 0.0002 ±0.0000 | -7.3094 ±0.0166 | 0.0181 ±0.0132 | 0.0006 ±0.0000 | 4.2194 ±0.0552 |
| GP-ARD-RBF | 0.0002 ±0.0000 | -7.1859 ±0.0146 | 0.0364 ±0.0021 | 0.0010 ±0.0000 | 14.0530 ±0.0552 |
| GP-ARD-Laplace | 0.0002 ±0.0000 | -7.3346 ±0.0094 | 0.0108 ±0.0031 | 0.0007 ±0.0000 | 15.4564 ±0.6676 |
| GP-ARD-Laplace-full | 0.0002 ±0.0000 | -6.7233 ±0.0045 | 0.0382 ±0.0016 | 0.0016 ±0.0000 | 9.4539 ±0.4092 |
| **GP-RFM-Laplace** | 0.0002 ±0.0000 | -7.3728 ±0.0094 | 0.0061 ±0.0031 | 0.0007 ±0.0000 | 16.4483 ±1.1083 |
| **GP-RFM-Laplace-diag** | 0.0002 ±0.0000 | -7.3641 ±0.0091 | 0.0073 ±0.0032 | 0.0007 ±0.0000 | 14.9629 ±0.2323 |
| NGBoost | 0.0002 ±0.0000 | -7.4229 ±0.0113 | 0.0042 ±0.0025 | 0.0006 ±0.0000 | 29.3339 ±0.1665 |
| CatBoost-Ensemble | 0.0002 ±0.0000 | -7.4136 ±0.0131 | 0.0071 ±0.0037 | 0.0006 ±0.0000 | 27.1441 ±2.6503 |

Table 10: OpenML dataset: houses (20640 samples; 8 covariates)

| | RMSE (↓) | NLL (↓) | CE (95%) (↓) | IL (95%) (↓) | Time (↓) |
|---|---|---|---|---|---|
| GP-RBF | 0.2555 ±0.0037 | 0.1620 ±0.0054 | 0.0346 ±0.0010 | 1.4874 ±0.0096 | 15.6545 ±0.2366 |
| GP-Laplace | 0.2528 ±0.0038 | 0.0727 ±0.0067 | 0.0216 ±0.0020 | 1.2362 ±0.0080 | 32.4636 ±1.2917 |
| deep Kernel Learning | 0.2647 ±0.0057 | 0.0919 ±0.0216 | 0.0061 ±0.0088 | 1.0388 ±0.0241 | 6.5323 ±0.0603 |
| GP-ARD-RBF | 0.2425 ±0.0038 | 0.1618 ±0.0059 | 0.0388 ±0.0012 | 1.5535 ±0.0107 | 15.7538 ±0.0603 |
| GP-ARD-Laplace | 0.2087 ±0.0035 | -0.0989 ±0.0064 | 0.0290 ±0.0020 | 1.1205 ±0.0086 | 32.3049 ±0.6497 |
| GP-ARD-Laplace-full | 0.2530 ±0.0034 | 0.1745 ±0.0049 | 0.0351 ±0.0020 | 1.5020 ±0.0041 | 19.3467 ±0.3817 |
| **GP-RFM-Laplace** | 0.2190 ±0.0032 | -0.0740 ±0.0066 | 0.0232 ±0.0020 | 1.0871 ±0.0087 | 34.3996 ±1.0498 |
| **GP-RFM-Laplace-diag** | 0.2245 ±0.0033 | -0.0377 ±0.0055 | 0.0257 ±0.0020 | 1.1525 ±0.0099 | 33.7937 ±0.2298 |
| NGBoost | 0.2826 ±0.0036 | 0.0747 ±0.0127 | 0.0041 ±0.0024 | 1.0640 ±0.0051 | 31.9747 ±0.2022 |
| CatBoost-Ensemble | 0.2345 ±0.0028 | -0.1249 ±0.0191 | 0.0387 ±0.0032 | 0.6976 ±0.0049 | 27.8312 ±2.6283 |

Table 11: OpenML dataset: house-16H (22784 samples; 16 covariates)

| | RMSE ($\downarrow$) | NLL ($\downarrow$) | CE (95%) ($\downarrow$) | IL (95%) ($\downarrow$) | Time ($\downarrow$) |
|---|---|---|---|---|---|
| GP-RBF | 0.6324 $_{\pm0.0281}$ | 0.8378 $_{\pm0.0294}$ | 0.0406 $_{\pm0.0016}$ | 2.8425 $_{\pm0.2764}$ | 20.8717 $_{\pm6.4480}$ |
| GP-Laplace | 0.6096 $_{\pm0.0274}$ | 0.7220 $_{\pm0.0241}$ | 0.0325 $_{\pm0.0022}$ | 2.2368 $_{\pm0.0525}$ | 39.2961 $_{\pm0.6178}$ |
| deep Kernel Learning | 0.6585 $_{\pm0.0551}$ | 0.9472 $_{\pm0.0505}$ | 0.0281 $_{\pm0.0137}$ | 2.4594 $_{\pm0.2951}$ | 7.4795 $_{\pm0.3042}$ |
| GP-ARD-RBF | 0.6352 $_{\pm0.0253}$ | 0.8382 $_{\pm0.0234}$ | 0.0408 $_{\pm0.0017}$ | 2.8555 $_{\pm0.2688}$ | 22.2921 $_{\pm0.3042}$ |
| GP-ARD-Laplace | 0.6077 $_{\pm0.0269}$ | 0.7171 $_{\pm0.0242}$ | 0.0328 $_{\pm0.0021}$ | 2.2158 $_{\pm0.0520}$ | 39.1299 $_{\pm0.1318}$ |
| GP-ARD-Laplace-full | 0.6168 $_{\pm0.0257}$ | 0.8972 $_{\pm0.0261}$ | 0.1304 $_{\pm0.2719}$ | 2.7582 $_{\pm0.0506}$ | 23.7313 $_{\pm0.4123}$ |
| **GP-RFM-Laplace** | 0.6063 $_{\pm0.0260}$ | 0.6899 $_{\pm0.0241}$ | 0.0318 $_{\pm0.0025}$ | 2.1696 $_{\pm0.0602}$ | 41.8046 $_{\pm0.9823}$ |
| **GP-RFM-Laplace-diag** | 0.6179 $_{\pm0.0283}$ | 0.7098 $_{\pm0.0315}$ | 0.0338 $_{\pm0.0024}$ | 2.2318 $_{\pm0.0699}$ | 41.4268 $_{\pm0.0609}$ |
| NGBoost | 0.6031 $_{\pm0.0280}$ | 0.5686 $_{\pm0.0467}$ | 0.0078 $_{\pm0.0029}$ | 1.5809 $_{\pm0.0292}$ | 72.4206 $_{\pm5.3819}$ |
| CatBoost-Ensemble | 0.5956 $_{\pm0.0308}$ | 0.5125 $_{\pm0.0586}$ | 0.0041 $_{\pm0.0035}$ | 1.4476 $_{\pm0.0891}$ | 32.0270 $_{\pm3.3470}$ |

Table 12: OpenML dataset: Brazilian-houses (10692 samples; 8 covariates)

| | RMSE ($\downarrow$) | NLL ($\downarrow$) | CE (95%) ($\downarrow$) | IL (95%) ($\downarrow$) | Time ($\downarrow$) |
|---|---|---|---|---|---|
| GP-RBF | 0.1049 $_{\pm0.0356}$ | -0.9903 $_{\pm0.6726}$ | 0.1723 $_{\pm0.2795}$ | 0.6436 $_{\pm0.4977}$ | 17.3515 $_{\pm6.7992}$ |
| GP-Laplace | 0.0622 $_{\pm0.0281}$ | -1.4862 $_{\pm0.0729}$ | 0.0451 $_{\pm0.0016}$ | 0.3345 $_{\pm0.0393}$ | 12.3126 $_{\pm0.5714}$ |
| deep Kernel Learning | 0.0493 $_{\pm0.0228}$ | -0.6394 $_{\pm0.0444}$ | 0.0841 $_{\pm0.1507}$ | 0.8156 $_{\pm0.0364}$ | 3.2681 $_{\pm0.0353}$ |
| GP-ARD-RBF | 0.0646 $_{\pm0.0318}$ | -2.1872 $_{\pm0.0338}$ | 0.0459 $_{\pm0.0017}$ | 0.1688 $_{\pm0.0076}$ | 9.7154 $_{\pm0.0353}$ |
| GP-ARD-Laplace | 0.0537 $_{\pm0.0284}$ | -1.8204 $_{\pm0.1340}$ | 0.0473 $_{\pm0.0011}$ | 0.2136 $_{\pm0.0055}$ | 12.4825 $_{\pm1.0629}$ |
| GP-ARD-Laplace-full | 0.0981 $_{\pm0.0216}$ | 0.2656 $_{\pm0.0432}$ | 0.0493 $_{\pm0.0004}$ | 2.0113 $_{\pm0.0849}$ | 6.8555 $_{\pm0.3227}$ |
| **GP-RFM-Laplace** | 0.0414 $_{\pm0.0181}$ | -2.1078 $_{\pm0.0600}$ | 0.0488 $_{\pm0.0005}$ | 0.1406 $_{\pm0.0049}$ | 12.8021 $_{\pm0.8225}$ |
| **GP-RFM-Laplace-diag** | 0.0404 $_{\pm0.0184}$ | -2.0733 $_{\pm0.0696}$ | 0.0486 $_{\pm0.0007}$ | 0.1566 $_{\pm0.0009}$ | 11.5711 $_{\pm0.5537}$ |
| NGBoost | 0.0529 $_{\pm0.0243}$ | -2.1812 $_{\pm0.1506}$ | 0.0194 $_{\pm0.0049}$ | 0.1061 $_{\pm0.0022}$ | 11.5133 $_{\pm0.5235}$ |
| CatBoost-Ensemble | 0.0541 $_{\pm0.0318}$ | -2.6618 $_{\pm0.2337}$ | 0.0398 $_{\pm0.0042}$ | 0.0833 $_{\pm0.0126}$ | 23.4218 $_{\pm2.4005}$ |

Table 13: OpenML dataset: Bike-Sharing-Demand (17379 samples; 6 covariates)

| | RMSE ($\downarrow$) | NLL ($\downarrow$) | CE (95%) ($\downarrow$) | IL (95%) ($\downarrow$) | Time ($\downarrow$) |
|---|---|---|---|---|---|
| GP-RBF | 110.0164 $_{\pm1.5813}$ | 6.1724 $_{\pm0.0069}$ | 0.0135 $_{\pm0.0026}$ | 544.7001 $_{\pm3.7165}$ | 12.4240 $_{\pm0.0800}$ |
| GP-Laplace | 108.5130 $_{\pm1.7637}$ | 6.1472 $_{\pm0.0096}$ | 0.0095 $_{\pm0.0037}$ | 520.2480 $_{\pm16.0873}$ | 22.4437 $_{\pm0.1403}$ |
| deep Kernel Learning | 103.8203 $_{\pm2.7178}$ | 6.0799 $_{\pm0.0479}$ | 0.0185 $_{\pm0.0093}$ | 445.8881 $_{\pm54.9197}$ | 5.5567 $_{\pm0.0227}$ |
| GP-ARD-RBF | 99.5496 $_{\pm1.1839}$ | 6.0465 $_{\pm0.0084}$ | 0.0089 $_{\pm0.0042}$ | 446.8315 $_{\pm9.7015}$ | 12.8033 $_{\pm0.0227}$ |
| GP-ARD-Laplace | 100.2501 $_{\pm1.1656}$ | 6.0335 $_{\pm0.0096}$ | 0.0176 $_{\pm0.0031}$ | 417.9397 $_{\pm2.5062}$ | 22.5115 $_{\pm0.2084}$ |
| GP-ARD-Laplace-full | 102.6751 $_{\pm1.4195}$ | 6.0678 $_{\pm0.0126}$ | 0.0047 $_{\pm0.0045}$ | 455.0795 $_{\pm11.7388}$ | 13.3893 $_{\pm0.1907}$ |
| **GP-RFM-Laplace** | 100.4792 $_{\pm1.2527}$ | 6.0351 $_{\pm0.0102}$ | 0.0192 $_{\pm0.0038}$ | 415.9705 $_{\pm4.3815}$ | 24.3658 $_{\pm1.6734}$ |
| **GP-RFM-Laplace-diag** | 100.4778 $_{\pm1.1564}$ | 6.0343 $_{\pm0.0097}$ | 0.0199 $_{\pm0.0041}$ | 414.1059 $_{\pm4.0362}$ | 23.5265 $_{\pm0.0726}$ |
| NGBoost | 104.1888 $_{\pm1.2904}$ | 5.6200 $_{\pm0.0110}$ | 0.0114 $_{\pm0.0028}$ | 337.1860 $_{\pm2.0218}$ | 12.8066 $_{\pm0.1632}$ |
| CatBoost-Ensemble | 100.3143 $_{\pm1.1865}$ | 5.5759 $_{\pm0.0120}$ | 0.0025 $_{\pm0.0022}$ | 310.3412 $_{\pm2.2997}$ | 20.1121 $_{\pm2.7261}$ |

Table 14: OpenML dataset: house-sales (21613 samples; 15 covariates)

| | RMSE (↓) | NLL (↓) | CE (95%) (↓) | IL (95%) (↓) | Time (↓) |
|---|---|---|---|---|---|
| GP-RBF | 0.2215 ±0.0051 | 0.0370 ±0.0182 | 0.0401 ±0.0019 | 1.3997 ±0.0306 | 24.7769 ±3.0601 |
| GP-Laplace | 0.2034 ±0.0027 | -0.1850 ±0.0078 | 0.0115 ±0.0027 | 0.9016 ±0.0043 | 35.0244 ±0.1595 |
| deep Kernel Learning | 0.1944 ±0.0034 | -0.2193 ±0.0168 | 0.0056 ±0.0044 | 0.7751 ±0.0265 | 6.8426 ±0.0507 |
| GP-ARD-RBF | 0.2028 ±0.0037 | -0.0021 ±0.0137 | 0.0416 ±0.0008 | 1.3650 ±0.0191 | 22.9284 ±0.0507 |
| GP-ARD-Laplace | 0.1808 ±0.0020 | -0.3033 ±0.0060 | 0.0127 ±0.0020 | 0.8016 ±0.0031 | 35.0499 ±0.2341 |
| GP-ARD-Laplace-full | 0.1995 ±0.0068 | -0.0552 ±0.1645 | 0.1210 ±0.2763 | 1.2464 ±0.3343 | 21.2724 ±0.5262 |
| **GP-RFM-Laplace** | 0.1755 ±0.0016 | -0.3151 ±0.0065 | 0.0177 ±0.0019 | 0.8292 ±0.0085 | 37.4970 ±0.5236 |
| **GP-RFM-Laplace-diag** | 0.1731 ±0.0023 | -0.3198 ±0.0100 | 0.0201 ±0.0026 | 0.8333 ±0.0174 | 37.1304 ±0.0767 |
| NGBoost | 0.2029 ±0.0020 | -0.2679 ±0.0083 | 0.0026 ±0.0015 | 0.7642 ±0.0033 | 39.2063 ±0.2366 |
| CatBoost-Ensemble | 0.1963 ±0.0021 | -0.3144 ±0.0085 | 0.0136 ±0.0090 | 0.6892 ±0.0028 | 29.7815 ±3.3777 |

Table 15: OpenML dataset: sulfur (10081 samples; 6 covariates)

| | RMSE (↓) | NLL (↓) | CE (95%) (↓) | IL (95%) (↓) | Time (↓) |
|---|---|---|---|---|---|
| GP-RBF | 0.0183 ±0.0027 | -2.4195 ±0.0201 | 0.0466 ±0.0012 | 0.1279 ±0.0043 | 7.6919 ±0.0673 |
| GP-Laplace | 0.0159 ±0.0031 | -2.8279 ±0.0344 | 0.0395 ±0.0019 | 0.0770 ±0.0027 | 12.2545 ±1.4937 |
| deep Kernel Learning | 0.0259 ±0.0038 | -2.4438 ±0.2366 | 0.0284 ±0.0112 | 0.0809 ±0.0091 | 3.3750 ±0.1205 |
| GP-ARD-RBF | 0.0182 ±0.0028 | -2.4195 ±0.0195 | 0.0467 ±0.0012 | 0.1280 ±0.0034 | 7.8406 ±0.1205 |
| GP-ARD-Laplace | 0.0169 ±0.0041 | -2.8275 ±0.0379 | 0.0394 ±0.0015 | 0.0775 ±0.0017 | 12.6240 ±1.4911 |
| GP-ARD-Laplace-full | 0.0182 ±0.0047 | -2.6309 ±0.1468 | 0.0418 ±0.0016 | 0.0952 ±0.0117 | 6.1771 ±0.1899 |
| **GP-RFM-Laplace** | 0.0171 ±0.0044 | -2.8000 ±0.0486 | 0.0395 ±0.0022 | 0.0781 ±0.0028 | 12.9883 ±1.2052 |
| **GP-RFM-Laplace-diag** | 0.0181 ±0.0043 | -2.7366 ±0.0560 | 0.0407 ±0.0018 | 0.0826 ±0.0032 | 10.8533 ±0.3308 |
| NGBoost | 0.0256 ±0.0042 | -2.5867 ±0.4146 | 0.0145 ±0.0141 | 0.0606 ±0.0053 | 11.3634 ±4.3508 |
| CatBoost-Ensemble | 0.0244 ±0.0046 | -2.8813 ±0.0822 | 0.0141 ±0.0162 | 0.0484 ±0.0050 | 25.1911 ±4.0039 |

Table 16: OpenML dataset: MiamiHousing2016 (13932 samples; 13 covariates)

| | RMSE (↓) | NLL (↓) | CE (95%) (↓) | IL (95%) (↓) | Time (↓) |
|---|---|---|---|---|---|
| GP-RBF | 0.1803 ±0.0047 | 0.0057 ±0.0041 | 0.0439 ±0.0011 | 1.4023 ±0.0078 | 10.2637 ±1.4351 |
| GP-Laplace | 0.1655 ±0.0041 | -0.3567 ±0.0117 | 0.0168 ±0.0036 | 0.8053 ±0.0073 | 15.4859 ±0.5375 |
| deep Kernel Learning | 0.1691 ±0.0048 | -0.3507 ±0.0236 | 0.0056 ±0.0043 | 0.7240 ±0.0214 | 4.3708 ±0.0464 |
| GP-ARD-RBF | 0.1785 ±0.0045 | 0.0044 ±0.0036 | 0.0441 ±0.0010 | 1.4056 ±0.0073 | 10.1812 ±0.0464 |
| GP-ARD-Laplace | 0.1486 ±0.0032 | -0.4600 ±0.0089 | 0.0216 ±0.0024 | 0.7449 ±0.0049 | 15.4611 ±0.6357 |
| GP-ARD-Laplace-full | 0.1670 ±0.0081 | -0.1694 ±0.1773 | 0.0348 ±0.0094 | 1.1301 ±0.2839 | 8.9669 ±0.3332 |
| **GP-RFM-Laplace** | 0.1485 ±0.0027 | -0.4744 ±0.0087 | 0.0178 ±0.0028 | 0.7233 ±0.0063 | 16.6506 ±0.7884 |
| **GP-RFM-Laplace-diag** | 0.1451 ±0.0027 | -0.5000 ±0.0085 | 0.0188 ±0.0025 | 0.7064 ±0.0063 | 15.3197 ±0.2462 |
| NGBoost | 0.1997 ±0.0038 | -0.3802 ±0.0143 | 0.0055 ±0.0037 | 0.7269 ±0.0061 | 38.6213 ±0.2251 |
| CatBoost-Ensemble | 0.1835 ±0.0039 | -0.5299 ±0.0164 | 0.0141 ±0.0190 | 0.5927 ±0.0063 | 30.2160 ±4.1676 |

Table 17: OpenML dataset: superconduct (21263 samples; 79 covariates)

| | RMSE (↓) | NLL (↓) | CE (95%) (↓) | IL (95%) (↓) | Time (↓) |
|---|---|---|---|---|---|
| GP-RBF | 11.9302 ±0.2640 | 4.2074 ±0.0045 | 0.0363 ±0.0014 | 91.1963 ±0.5064 | 15.8843 ±0.0842 |
| GP-Laplace | 9.4970 ±0.2163 | 4.0099 ±0.0127 | 0.0281 ±0.0017 | 72.7501 ±0.7574 | 34.5985 ±0.6591 |
| deep Kernel Learning | 14.5717 ±0.4363 | 4.0989 ±0.0298 | 0.0097 ±0.0029 | 58.0927 ±1.2015 | 6.7805 ±0.0431 |
| GP-ARD-RBF | 11.8767 ±0.2608 | 4.2020 ±0.0046 | 0.0364 ±0.0017 | 90.8332 ±0.5023 | 17.0431 ±0.0431 |
| GP-ARD-Laplace | 9.5881 ±0.2008 | 3.9596 ±0.0106 | 0.0291 ±0.0020 | 68.9498 ±0.6899 | 34.4697 ±0.5762 |
| GP-ARD-Laplace-full | 11.2755 ±0.8971 | 4.3354 ±0.1772 | 0.0887 ±0.1950 | 111.7846 ±21.7467 | 28.8458 ±33.2999 |
| **GP-RFM-Laplace** | 13.2086 ±7.0422 | 4.0457 ±0.1059 | 0.0409 ±0.0362 | 93.3640 ±34.0538 | 71.6949 ±83.6162 |
| **GP-RFM-Laplace-diag** | 10.3174 ±0.2215 | 4.0254 ±0.0094 | 0.0294 ±0.0022 | 75.3887 ±1.3068 | 37.3528 ±0.9843 |
| NGBoost | 13.2014 ±0.1743 | 3.6477 ±0.0177 | 0.0064 ±0.0028 | 43.5723 ±0.3922 | 262.9271 ±1.4310 |
| CatBoost-Ensemble | 10.9449 ±1.0811 | 3.4594 ±0.1136 | 0.0236 ±0.0174 | 29.7679 ±6.6060 | 61.2334 ±5.3609 |

Table 18: OpenML dataset: california (20640 samples; 8 covariates)

| | RMSE (↓) | NLL (↓) | CE (95%) (↓) | IL (95%) (↓) | Time (↓) |
|---|---|---|---|---|---|
| GP-RBF | 0.1614 ±0.0025 | -0.3129 ±0.0082 | 0.0325 ±0.0013 | 0.9172 ±0.0064 | 16.3839 ±0.2506 |
| GP-Laplace | 0.1591 ±0.0026 | -0.3960 ±0.0148 | 0.0201 ±0.0017 | 0.7693 ±0.0112 | 32.4422 ±0.8164 |
| deep Kernel Learning | 0.1645 ±0.0039 | -0.3389 ±0.0859 | 0.0144 ±0.0147 | 0.7538 ±0.1859 | 6.4947 ±0.1318 |
| GP-ARD-RBF | 0.1478 ±0.0016 | -0.3194 ±0.0052 | 0.0389 ±0.0011 | 0.9708 ±0.0049 | 16.1321 ±0.1318 |
| GP-ARD-Laplace | 0.1233 ±0.0020 | -0.6350 ±0.0067 | 0.0246 ±0.0022 | 0.6407 ±0.0045 | 32.3389 ±0.9311 |
| GP-ARD-Laplace-full | 0.1528 ±0.0024 | -0.4045 ±0.0317 | 0.0242 ±0.0063 | 0.7753 ±0.0627 | 19.4896 ±0.1069 |
| **GP-RFM-Laplace** | 0.1297 ±0.0018 | -0.5954 ±0.0056 | 0.0224 ±0.0021 | 0.6503 ±0.0048 | 34.3716 ±0.7675 |
| **GP-RFM-Laplace-diag** | 0.1261 ±0.0017 | -0.6142 ±0.0052 | 0.0242 ±0.0016 | 0.6513 ±0.0047 | 33.9779 ±0.4784 |
| NGBoost | 0.1674 ±0.0019 | -0.4479 ±0.0110 | 0.0027 ±0.0015 | 0.6248 ±0.0044 | 35.9571 ±0.2644 |
| CatBoost-Ensemble | 0.1443 ±0.0017 | -0.5852 ±0.0235 | 0.0396 ±0.0040 | 0.4275 ±0.0043 | 28.9227 ±3.3222 |

Table 19: OpenML dataset: fifa (18063 samples; 5 covariates)

| | RMSE (↓) | NLL (↓) | CE (95%) (↓) | IL (95%) (↓) | Time (↓) |
|---|---|---|---|---|---|
| GP-RBF | 0.8447 ±0.0112 | 1.2718 ±0.0083 | 0.0194 ±0.0032 | 4.1114 ±0.0515 | 13.9853 ±0.6240 |
| GP-Laplace | 0.8367 ±0.0106 | 1.2434 ±0.0092 | 0.0132 ±0.0023 | 3.6669 ±0.0112 | 24.2460 ±0.1180 |
| deep Kernel Learning | 0.7928 ±0.0096 | 1.1871 ±0.0126 | 0.0059 ±0.0029 | 3.0820 ±0.0194 | 5.5823 ±0.0410 |
| GP-ARD-RBF | 0.8226 ±0.0091 | 1.2295 ±0.0068 | 0.0108 ±0.0027 | 3.8480 ±0.0312 | 14.5917 ±0.0410 |
| GP-ARD-Laplace | 0.8203 ±0.0102 | 1.2215 ±0.0091 | 0.0131 ±0.0043 | 3.7242 ±0.1273 | 24.4010 ±0.2973 |
| GP-ARD-Laplace-full | 0.8060 ±0.0097 | 1.1942 ±0.0095 | 0.0026 ±0.0022 | 3.3354 ±0.0318 | 14.5039 ±0.0771 |
| **GP-RFM-Laplace** | 0.8093 ±0.0135 | 1.2056 ±0.0194 | 0.0068 ±0.0049 | 3.5477 ±0.2081 | 26.2624 ±1.3755 |
| **GP-RFM-Laplace-diag** | 0.7982 ±0.0101 | 1.1864 ±0.0092 | 0.0043 ±0.0042 | 3.3570 ±0.0129 | 25.7180 ±0.0734 |
| NGBoost | 0.7746 ±0.0096 | 1.0939 ±0.0116 | 0.0136 ±0.0030 | 2.8662 ±0.0161 | 14.2918 ±0.6475 |
| CatBoost-Ensemble | 0.7768 ±0.0101 | 1.0964 ±0.0178 | 0.0147 ±0.0033 | 2.8305 ±0.0502 | 25.0984 ±3.5605 |

## C.2 UCI BENCHMARK

Table 20: UCI dataset: Concrete Compression Strength (1030 samples; 8 covariates)

| | RMSE (↓) | NLL (↓) | CE (95%) (↓) | IL (95%) (↓) | Time (↓) |
|---|---|---|---|---|---|
| GP-RBF | 5.1649 ±0.8539 | 3.3152 ±0.0541 | 0.0417 ±0.0098 | 39.9684 ±2.1881 | 5.6247 ±0.1457 |
| GP-Laplace | 4.9342 ±0.6294 | 3.1955 ±0.1154 | 0.0350 ±0.0104 | 30.6059 ±0.8027 | 4.2635 ±0.3242 |
| deep Kernel Learning | 5.9734 ±0.6219 | 3.2103 ±0.1078 | 0.0257 ±0.0266 | 23.3185 ±1.2344 | 1.5215 ±0.0246 |
| GP-ARD-RBF | 5.0771 ±0.8361 | 3.2424 ±0.0776 | 0.0413 ±0.0097 | 36.5128 ±2.0837 | 4.4744 ±0.0246 |
| GP-ARD-Laplace | 4.8375 ±0.7156 | 3.0636 ±0.1369 | 0.0261 ±0.0156 | 25.1250 ±0.6902 | 4.2398 ±0.4143 |
| GP-ARD-Laplace-full | 5.8577 ±0.5434 | 3.5410 ±0.0146 | 0.0471 ±0.0054 | 48.5448 ±0.3850 | 4.8013 ±0.6353 |
| **GP-RFM-Laplace** | 4.9390 ±0.6834 | 3.0209 ±0.0878 | 0.0243 ±0.0157 | 24.1509 ±0.8528 | 4.4505 ±0.4256 |
| **GP-RFM-Laplace-diag** | 5.3889 ±0.8233 | 3.0977 ±0.1039 | 0.0199 ±0.0133 | 25.2850 ±0.6942 | 4.3315 ±0.1873 |
| NGBoost | 5.6672 ±0.6433 | 3.0846 ±0.1400 | 0.0298 ±0.0280 | 18.9717 ±1.2642 | 3.1179 ±0.3985 |
| CatBoost-Ensemble | 5.3957 ±0.5575 | 3.0866 ±0.1788 | 0.0414 ±0.0344 | 17.0637 ±1.8528 | 13.1103 ±0.9603 |

Table 21: UCI dataset: Energy Efficiency (768 samples; 8 covariates)

| | RMSE (↓) | NLL (↓) | CE (95%) (↓) | IL (95%) (↓) | Time (↓) |
|---|---|---|---|---|---|
| GP-RBF | 0.5372 ±0.0881 | 0.8400 ±0.0745 | 0.0338 ±0.0142 | 2.8408 ±0.0700 | 0.8882 ±0.0090 |
| GP-Laplace | 1.6181 ±0.1648 | 1.9621 ±0.0585 | 0.0279 ±0.0160 | 8.3290 ±0.0656 | 1.3786 ±0.0936 |
| deep Kernel Learning | 0.6141 ±0.1244 | 1.9590 ±0.0368 | 0.0500 ±0.0000 | 10.8165 ±0.3824 | 0.5040 ±0.0166 |
| GP-ARD-RBF | 0.4482 ±0.0466 | 0.6581 ±0.0641 | 0.0249 ±0.0108 | 2.2212 ±0.0304 | 0.9271 ±0.0166 |
| GP-ARD-Laplace | 0.5804 ±0.0655 | 1.1556 ±0.0268 | 0.0481 ±0.0046 | 4.3960 ±0.0471 | 1.5367 ±0.3320 |
| GP-ARD-Laplace-full | 2.6116 ±0.2181 | 2.9738 ±0.0099 | 0.0500 ±0.0000 | 28.6954 ±0.1360 | 0.7235 ±0.0093 |
| **GP-RFM-Laplace** | 0.4965 ±0.0431 | 0.9515 ±0.0279 | 0.0481 ±0.0046 | 3.4807 ±0.0345 | 1.3571 ±0.2191 |
| **GP-RFM-Laplace-diag** | 0.4849 ±0.0441 | 0.8953 ±0.0309 | 0.0474 ±0.0052 | 3.2305 ±0.0402 | 1.0165 ±0.0221 |
| NGBoost | 0.5141 ±0.0463 | 0.6078 ±0.1680 | 0.0274 ±0.0148 | 1.9866 ±0.2474 | 2.7234 ±0.2241 |
| CatBoost-Ensemble | 0.5588 ±0.1193 | 0.5794 ±0.2291 | 0.0373 ±0.0203 | 1.7964 ±0.6647 | 5.9525 ±0.7011 |

Table 22: UCI dataset: Kin8nm (8192 samples; 8 covariates)

| | RMSE (↓) | NLL (↓) | CE (95%) (↓) | IL (95%) (↓) | Time (↓) |
|---|---|---|---|---|---|
| GP-RBF | 0.0756 ±0.0023 | -0.7730 ±0.0057 | 0.0499 ±0.0003 | 0.6547 ±0.0019 | 7.8375 ±0.0981 |
| GP-Laplace | 0.0761 ±0.0025 | -1.0735 ±0.0148 | 0.0376 ±0.0033 | 0.4172 ±0.0014 | 12.5205 ±0.5642 |
| deep Kernel Learning | 0.0722 ±0.0058 | -1.1991 ±0.0704 | 0.0157 ±0.0082 | 0.3139 ±0.0172 | 3.3887 ±0.0272 |
| GP-ARD-RBF | 0.0747 ±0.0023 | -0.7759 ±0.0056 | 0.0499 ±0.0003 | 0.6541 ±0.0019 | 7.8674 ±0.0272 |
| GP-ARD-Laplace | 0.0722 ±0.0022 | -1.1377 ±0.0147 | 0.0381 ±0.0036 | 0.3853 ±0.0012 | 12.0966 ±0.4330 |
| GP-ARD-Laplace-full | 0.0833 ±0.0062 | -0.8490 ±0.1591 | 0.0457 ±0.0044 | 0.5756 ±0.1107 | 7.4953 ±0.3022 |
| **GP-RFM-Laplace** | 0.0657 ±0.0018 | -1.2620 ±0.0109 | 0.0321 ±0.0042 | 0.3282 ±0.0026 | 13.2497 ±0.6537 |
| **GP-RFM-Laplace-diag** | 0.0755 ±0.0016 | -1.0997 ±0.0106 | 0.0381 ±0.0036 | 0.3948 ±0.0012 | 11.9409 ±0.1580 |
| NGBoost | 0.1819 ±0.0038 | -0.3626 ±0.0185 | 0.0082 ±0.0053 | 0.6536 ±0.0051 | 22.3981 ±0.2512 |
| CatBoost-Ensemble | 0.1388 ±0.0032 | -0.6557 ±0.0219 | 0.0081 ±0.0051 | 0.4712 ±0.0029 | 24.8655 ±1.6878 |

Table 23: UCI dataset: Naval Plant Maintenance (11934 samples; 16 covariates)

| | RMSE (↓) | NLL (↓) | CE (95%) (↓) | IL (95%) (↓) | Time (↓) |
|---|---|---|---|---|---|
| GP-RBF | 0.0002 ±0.0001 | -6.1782 ±0.8216 | 0.1608 ±0.2245 | 0.0023 ±0.0000 | 11.5056 ±1.2598 |
| GP-Laplace | 0.0003 ±0.0000 | -5.7682 ±0.0052 | 0.0500 ±0.0002 | 0.0049 ±0.0000 | 17.4098 ±0.2460 |
| deep Kernel Learning | 0.0005 ±0.0001 | -4.1528 ±0.2374 | 0.0500 ±0.0000 | 0.0253 ±0.0070 | 4.6008 ±0.0534 |
| GP-ARD-RBF | 0.0002 ±0.0000 | -6.4786 ±0.0103 | 0.1825 ±0.3225 | 0.0023 ±0.0000 | 10.9200 ±0.0534 |
| GP-ARD-Laplace | 0.0002 ±0.0000 | -5.9669 ±0.0043 | 0.0500 ±0.0000 | 0.0040 ±0.0000 | 17.2876 ±0.1934 |
| GP-ARD-Laplace-full | 0.0005 ±0.0001 | -4.1346 ±0.2305 | 0.0500 ±0.0000 | 0.0255 ±0.0047 | 10.4092 ±0.4880 |
| **GP-RFM-Laplace** | 0.0003 ±0.0000 | -5.9195 ±0.0361 | 0.0500 ±0.0000 | 0.0041 ±0.0002 | 19.8079 ±5.2316 |
| **GP-RFM-Laplace-diag** | 0.0002 ±0.0000 | -5.8776 ±0.0038 | 0.0500 ±0.0000 | 0.0044 ±0.0000 | 18.3550 ±0.0997 |
| NGBoost | 0.0059 ±0.0001 | -3.9178 ±0.0138 | 0.0479 ±0.0013 | 0.0234 ±0.0002 | 37.5817 ±5.8329 |
| CatBoost-Ensemble | 0.0016 ±0.0001 | -5.4828 ±0.0322 | 0.0444 ±0.0023 | 0.0059 ±0.0002 | 29.2242 ±2.6337 |

Table 24: UCI dataset: Combined Cycle Power Plant (9568 samples; 4 covariates)

| | RMSE (↓) | NLL (↓) | CE (95%) (↓) | IL (95%) (↓) | Time (↓) |
|---|---|---|---|---|---|
| GP-RBF | 3.9517 ±0.1233 | 3.1728 ±0.0070 | 0.0484 ±0.0010 | 33.5592 ±0.0525 | 9.2673 ±0.8142 |
| GP-Laplace | 3.4041 ±0.1247 | 2.7214 ±0.0198 | 0.0310 ±0.0037 | 17.8956 ±0.0945 | 14.7721 ±0.7598 |
| deep Kernel Learning | 3.9615 ±0.1201 | 2.8441 ±0.0209 | 0.0362 ±0.0040 | 19.6622 ±0.5423 | 3.7840 ±0.0196 |
| GP-ARD-RBF | 3.9448 ±0.1227 | 3.1725 ±0.0069 | 0.0484 ±0.0010 | 33.5615 ±0.0524 | 9.2302 ±0.0196 |
| GP-ARD-Laplace | 2.6824 ±0.1770 | 2.7131 ±0.0176 | 0.0438 ±0.0023 | 20.8760 ±0.1848 | 15.0723 ±0.6198 |
| GP-ARD-Laplace-full | 3.5091 ±0.2767 | 2.8893 ±0.2700 | 0.0824 ±0.1988 | 23.0570 ±10.6857 | 11.6361 ±15.4711 |
| **GP-RFM-Laplace** | 3.2018 ±0.1364 | 2.6849 ±0.0214 | 0.0343 ±0.0034 | 17.7328 ±0.1058 | 15.2813 ±0.5343 |
| **GP-RFM-Laplace-diag** | 3.2550 ±0.1277 | 2.6963 ±0.0200 | 0.0342 ±0.0039 | 17.8465 ±0.0886 | 14.0767 ±0.1701 |
| NGBoost | 3.8750 ±0.1488 | 2.7677 ±0.0749 | 0.0074 ±0.0041 | 14.5355 ±0.6960 | 11.9584 ±2.2816 |
| CatBoost-Ensemble | 3.3635 ±0.3802 | 2.6596 ±0.1451 | 0.0220 ±0.0156 | 11.1089 ±1.9365 | 25.3693 ±4.5353 |

Table 25: UCI dataset: Wine Quality Red (1599 samples; 11 covariates)

| | RMSE (↓) | NLL (↓) | CE (95%) (↓) | IL (95%) (↓) | Time (↓) |
|---|---|---|---|---|---|
| GP-RBF | 0.6026 ±0.0440 | 0.9114 ±0.0648 | 0.0194 ±0.0099 | 2.4992 ±0.0667 | 3.2311 ±0.0717 |
| GP-Laplace | 0.5517 ±0.0433 | 0.8475 ±0.0521 | 0.0197 ±0.0095 | 2.6585 ±0.0346 | 4.1511 ±0.1734 |
| deep Kernel Learning | 0.6440 ±0.0587 | 0.9905 ±0.1014 | 0.0266 ±0.0190 | 2.4026 ±0.1845 | 1.4830 ±0.0280 |
| GP-ARD-RBF | 0.6033 ±0.0412 | 0.9202 ±0.0553 | 0.0172 ±0.0095 | 2.6270 ±0.1211 | 3.3238 ±0.0280 |
| GP-ARD-Laplace | 0.5635 ±0.0435 | 0.8547 ±0.0528 | 0.0181 ±0.0117 | 2.6115 ±0.0285 | 3.9449 ±0.2618 |
| GP-ARD-Laplace-full | 0.5747 ±0.0416 | 0.8811 ±0.0519 | 0.0203 ±0.0115 | 2.6116 ±0.0423 | 2.3756 ±0.2144 |
| **GP-RFM-Laplace** | 0.5569 ±0.0435 | 0.8500 ±0.0558 | 0.0184 ±0.0094 | 2.6239 ±0.0531 | 4.0593 ±0.2478 |
| **GP-RFM-Laplace-diag** | 0.5617 ±0.0488 | 0.8618 ±0.0649 | 0.0200 ±0.0126 | 2.6260 ±0.0683 | 4.0605 ±0.4318 |
| NGBoost | 0.6212 ±0.0445 | 0.9278 ±0.0741 | 0.0209 ±0.0132 | 2.2635 ±0.1561 | 2.6412 ±0.6679 |
| CatBoost-Ensemble | 0.6134 ±0.0507 | 0.8973 ±0.1082 | 0.0291 ±0.0208 | 2.0483 ±0.0399 | 15.4885 ±2.2552 |

Table 26: UCI dataset: Yacht Hydrodynamics (308 samples; 6 covariates)

| | RMSE (↓) | NLL (↓) | CE (95%) (↓) | IL (95%) (↓) | Time (↓) |
|---|---|---|---|---|---|
| GP-RBF | 0.8173 ±0.2675 | 1.0579 ±0.0704 | 0.0366 ±0.0164 | 3.7188 ±0.1813 | 0.7283 ±0.0648 |
| GP-Laplace | 2.9169 ±0.8372 | 2.5045 ±0.1010 | 0.0310 ±0.0190 | 15.6999 ±0.3866 | 1.1829 ±0.2183 |
| deep Kernel Learning | 0.7553 ±0.3034 | 2.3936 ±0.0231 | 0.0500 ±0.0000 | 16.8325 ±0.4511 | 0.4360 ±0.0097 |
| GP-ARD-RBF | 0.5183 ±0.2543 | 0.8530 ±0.2459 | 0.0368 ±0.0159 | 2.8902 ±0.0785 | 0.8103 ±0.0097 |
| GP-ARD-Laplace | 1.1895 ±0.5616 | 1.9742 ±0.0777 | 0.0366 ±0.0164 | 10.1187 ±0.1586 | 1.4123 ±0.3088 |
| GP-ARD-Laplace-full | 4.6409 ±1.4430 | 3.4261 ±0.0448 | 0.0468 ±0.0097 | 44.0111 ±0.5874 | 0.7122 ±0.0339 |
| **GP-RFM-Laplace** | 1.0447 ±0.3585 | 1.4985 ±0.2114 | 0.0323 ±0.0162 | 4.7886 ±0.1456 | 1.2185 ±0.2006 |
| **GP-RFM-Laplace-diag** | 1.0187 ±0.3369 | 1.5235 ±0.1761 | 0.0310 ±0.0169 | 5.2000 ±0.2562 | 0.8918 ±0.0676 |
| NGBoost | 0.7487 ±0.2946 | 0.7046 ±0.4721 | 0.0402 ±0.0364 | 2.1121 ±0.9981 | 2.0757 ±0.2763 |
| CatBoost-Ensemble | 1.2838 ±0.7056 | 0.4008 ±0.6086 | 0.1053 ±0.0749 | 1.5808 ±1.3508 | 5.0563 ±1.0566 |