# OpenReview forum: "Uncertainty Estimation with Recursive Feature Machines"
_auai.org/UAI/2024/Conference — UAI 2024 poster_

### Official Review · Reviewer_BycZ · 2024-03-21

**Q2-1 Originality-Novelty:** 2
**Q2-2 Correctness-Technical Quality:** 3
**Q2-5 Clarity Of Writing:** 3

**Q10 Ethical Concerns:**

No.

**Q1 Summary And Contributions:**

This paper addresses the importance of uncertainty quantification in predictive modeling. It introduces the Recursive Feature Machine (RFM) as a data-adaptive kernel-based method for estimating uncertainty. On the other hand, gaussian processes (GPs) are powerful non-parametric models which offer uncertainty quantification, however, they are limited by their reliance on kernel functions that are not adaptive to data. So to fill in this gap; they adopt the RFM as a novel data-adaptive, feature learning kernel for uncertainty quantification through integration into GPs.

**Q2-3 Extent To Which Claims Are Supported By Evidence:**

2: Fair: the main claims are somewhat supported by evidence (but the experimental evaluation may be weak, or does not match entirely with the claims, important baselines may be missing, proofs contain important ideas but lack rigor, algorithmic details are only discussed superficially, references are imprecise, assumptions are not sufficiently motivated or explicated, etc.).

**Q2-4 Reproducibility:**

3: Good: key resources (e.g. proofs, code, data) are available and key details (e.g. proofs, experimental setup) are sufficiently well-described for competent researchers to confidently reproduce the main results.

**Q3 Main Strengths:**

1. The idea of integrating RFM with GP is interesting and worth exploring.
2. The experimental settings seem technically sound to me.
3. They showed the performance of the proposed model on different benchmark studies, and synthetic data.

**Q4 Main Weakness:**

Some clarification might be needed in experiments.

**Q5 Detailed Comments To The Authors:**

1. That would be nicer, if the authors could add pseudo code/algorithm for better understanding of the readers.
2. In generation toy data set, it is mentioned that the covariates are independent, at the same time, they claimed that this dataset is designed to be challenging for the methods which do not consider covariate correlation. These two sentences sound inconsistent to me! Also in visualizing feature matrices, they mentioned that we generated the labels with correlating covariates according to (11). Are these covariates are generated to be dependent at the end?
3. The performance of the model has been shown on different benchmark studies and in comparison with baseline models, however I am wondering why in the toy data set, the proposed models are not compared to the baseline models as well?
4. In some of the tables from the appendix, the MSE of baseline model is lower than the proposed model, and the computational time of the proposed model is higher compared to baseline models, for instance table 16 (GP-ARD-Laplace vs GP-RFM-Laplace). What is happening for this data set, or I missed something here?
5. Some notations are not clearly explained in different equations for instance in equation 2, what does i indicate, or lambda? or in section 5.3, what does jth dimension of ith sample mean? it is not clear to me.

**Q9 Complying With Reviewing Instructions:**

Yes

---

> ### Author Rebuttal · Authors · 2024-04-05
>
> We thank the reviewer for the positive feedback. Below, we respond to the questions raised.
>
> 1. We will add the pseudo code to the final version of the paper.
> 2. Thank you for pointing this out, we abused the term correlation here and we will replace it. What we meant was that the prediction $y$ is dependent on a **combination** of different covariates $x$.
> 3. In Figure 3(main body) and Figure 6 (in the Appendix) we compare with baseline methods for the toy dataset. Please let us know if we missed anything.
> 4. We acknowledge that it is important to not overstate our findings. So far, we do not claim that our method is always superior to state-of-the-art methods but rather presents a highly competitive, new approach. We already stated so in the discussion but will emphasise this more clearly.
> 5. Thanks for pointing this out, we will add the definition of ridge $\lambda$ in equation (2). In section 5.3, we mention in the following line that "$x_{i[j]}$ denotes the $j$th dimension of the $i$th sample". Please let us know if it is still unclear.

---

### Official Review · Reviewer_iQjj · 2024-03-22

**Q2-1 Originality-Novelty:** 3
**Q2-2 Correctness-Technical Quality:** 3
**Q2-5 Clarity Of Writing:** 3

**Q1 Summary And Contributions:**

This paper introduces a new method of uncertainty estimation in regression analysis.  The method is based on the recently proposed Recursive Feature Machines (RFMs) that is aimed towards data-adaptive feature-learning kernel, to improve classical Gaussian Process kernel-based frameworks. The RFM algorithm (Radhakrishnan et. al.) learns a feature matrix from data using the idea of Average Gradient Outer Product (AGOP). In this paper the authors integrated the RFMs into GPs by replacing the standard kernels (Laplace) with RFM based kernels. The resulting method is called GP-RFM-Laplace. The authors claim that GP-RFM-Laplace outperforms other leading uncertainty estimation techniques in regression analysis involving tabular and categorical data.  These include traditional GP based techniques such as GP-RBF, GP-ARD etc., and boosting-based approaches such as NGBoost and CatBoost-Ensemble.

**Q2-3 Extent To Which Claims Are Supported By Evidence:**

3: Good: the main claims are supported by convincing evidence (in the form of adequate experimental evaluation, proofs, (pseudo-)code, references, assumptions).

**Q2-4 Reproducibility:**

2: Fair: key resources (e.g. proofs, code, data) are unavailable but key details (e.g. proof sketches, experimental setup) are sufficiently well-described for an expert to confidently reproduce the main results.

**Q3 Main Strengths:**

•	The writing is clear and unambiguous. The techniques, old and new, are laid out clearly with sufficient mathematical details to serve as a sampler.
•	Details about the implementation are provided (source of code and datasets)
•	The evaluation results are arrived at through experimenting with sufficiently diverse methods and datasets (within the scope of the claims of the paper)

**Q4 Main Weakness:**

•	Lacks a more detailed theoretical justification of why integrating RFMs reinforces GPs the way they do (beyond the high-level understanding that RFMs encode covariate correlations)

**Q5 Detailed Comments To The Authors:**

the article is well written. To improve please refer to the Q4 comment for details. Please also share the prototypical implementation to the audience as promised.

**Q9 Complying With Reviewing Instructions:**

Yes

---

> ### Author Rebuttal · Authors · 2024-04-05
>
> We thank the reviewer for the positive feedback. The theoretical analysis of our method is indeed an interesting future work. However, it is beyond the current scope of this paper. We will release the GitHub code upon acceptance and we plan to add the pseudo code to the final version of the paper.

---

### Official Review · Reviewer_Fi6K · 2024-03-23

**Q2-1 Originality-Novelty:** 2
**Q2-2 Correctness-Technical Quality:** 3
**Q2-5 Clarity Of Writing:** 3

**Q1 Summary And Contributions:**

The authors propose a kernel regression method (recursive feature machines) for Gaussian processes for use on tabular regression data. RFMs parametrize the kernel through a distance/covariance-adjusted Mahalanobis distance between data points. Detailed results are presented on OpenML and UCI regression benchmarks against other common tabular data baselines.

**Q2-3 Extent To Which Claims Are Supported By Evidence:**

2: Fair: the main claims are somewhat supported by evidence (but the experimental evaluation may be weak, or does not match entirely with the claims, important baselines may be missing, proofs contain important ideas but lack rigor, algorithmic details are only discussed superficially, references are imprecise, assumptions are not sufficiently motivated or explicated, etc.).

**Q2-4 Reproducibility:**

2: Fair: key resources (e.g. proofs, code, data) are unavailable but key details (e.g. proof sketches, experimental setup) are sufficiently well-described for an expert to confidently reproduce the main results.

**Q3 Main Strengths:**

The main strengths of the paper lie in the clear flow and language of the paper, as well as the extensiveness of experimental results presented in the main text and supplement. I especially appreciated that detailed metrics are provided for every baseline and experiment in numerous UCI and OpenML benchmarks.

**Q4 Main Weakness:**

The main weakness of the paper is the lack of clarity in regards to the novelty and theoretical details of the RFM, as well as mixed experimental results/presentation. Please correct me if I am wrong, but so far as I can tell the RFM has been presented so far only in pre-prints, and as such there is a higher burden of presentation of the RFM method and theoretical justification than is given here. In addition, it seems the experimental results are somewhat mixed in both NLL and RMSE performance (see Q5 response), which amplifies the issue of lack of theoretical contributions. I think the paper would benefit from more specific focus on one of these two components; ie. clear theoretical presentation or clear improvement on baselines.

**Q5 Detailed Comments To The Authors:**

Lack of Theoretical Results:
- Again, please correct me if I am wrong, but I don't think that the current presentation of RFM is sufficient given that it has not been peer-reviewed/published prior. I think it requires a deeper presentation here as a result in order to be considered for publication. For example - how is RFM different than an RFF approach which can, in theory, model any stationary kernel? What are the competitive advantages over something like a deep kernel which is a universal approximator? Specifically, I think it should be presented why the authors sought alternatives to these methods and theoretically (not experimentally) justify their method.
- Once those justifications have been made, I think it is important to then give a full presentation of the RFM methodology.

Mixed Experimental Results:
- The NLL results are mixed, with the author's method yielding the best performance on a handful of datasets.
- For the RMSE results, the authors claim based on figure 2 that their methods yield superior RMSE performance, but I found this claim somewhat confusing. I think the method of normalizing and aggregating RMSE results across datasets is not particularly common/representative as a metric. Some datasets will have (even when normalized) naturally lower RMSE, and this way of presenting baselines just favors those models which performed well on those datasets. By my count based on the tables in the appendices for the OpenML benchmarks, their methods are not top performing on at least 9 out of the 16 datasets (please correct if wrong).
- As such, I think it is more clear and better to just include a summary table for RMSE across datasets rather than figure 2.

Misc:
- I would recommend you put in 1-2 lines describing RFM's into the intro.
- I would recommend prior work section go a little bit deeper - I assume most readers will in general be familiar with these concepts - I would explore more kernel learning settings and direct performance comparisons between GBM and GPs to justify the statement that they outperform GPs. Maybe just move the related works to end as I think it's probably unnecessary in its current state for most readers.
- "integrates a positive semi-definite, symmetric matrix "-> integrates may not be the right word here given the dual mathematical definition.
- The detailed appendix is appreciated, but I think RMSE + calibration is more useful set of metrics to present in the main body.

**Q9 Complying With Reviewing Instructions:**

Yes

---

> ### Author Rebuttal · Authors · 2024-04-05
>
> We thank the reviewer for the feedback. In what follows we address the concerns raised,
>
> 1. *Please correct me if I am wrong, but so far as I can tell the RFM has been presented so far only in pre-prints, and as such there is a higher burden of presentation of the RFM method and theoretical justification than is given here.* and *Again, please correct me if I am wrong, but I don't think that the current presentation of RFM is sufficient given that it has not been peer-reviewed/published prior. I think it requires a deeper presentation here as a result in order to be considered for publication.*
>
> The paper on RFM was recently published in Science and can be accessed here: https://www.science.org/doi/abs/10.1126/science.adi5639.
> This paper provides an extensive examination of the concept which is why we concentrate on a summary of the method. If you have any specific suggestions we would be happy to incorporate that in the final version of the paper.
>
> 2. *For example - how is RFM different than an RFF approach which can, in theory, model any stationary kernel? What are the competitive advantages over something like a deep kernel which is a universal approximator? Specifically, I think it should be presented why the authors sought alternatives to these methods and theoretically (not experimentally) justify their method.*
>
> While they are both universal approximates, RFM helps with the sample complexity (a large reduction in sample complexity). Moreover, exploring the theoretical underpinnings of our method presents a compelling opportunity for future research. However, it falls beyond the scope of this paper and constitutes an intriguing area for future investigation.
>
> 3. *I think the method of normalizing and aggregating RMSE results across datasets is not particularly common/representative as a metric. Some datasets will have (even when normalized) naturally lower RMSE, and this way of presenting baselines just favors those models which performed well on those datasets.*
>
> We thank the reviewer for pointing this out, this is indeed a valid concern. However, based on the Tables in Appendix C, it can be seen this is not the case that the RMSE are very far from each other. We will comment on this and add a summary Table for RMSE similar to Tables 1 and 2 as you suggested in the final version of the paper.
>
> 4. *The NLL results are mixed, with the author's method yielding the best performance on a handful of datasets.*
>
> Our method demonstrates competitive performance across all datasets and achieves the best results on several. While we do not claim it universally outperforms all others, as stated in our paper, we acknowledge that it is important not to overstate our findings and therefore emphasise this more clearly from early on. However, it is uncommon for a single algorithm to excel across every dataset. Thus, we think our contribution is a new algorithm with very strong performance.
>
> 5. *Misc:*
> Thank you for the misc feedback. We will rectify them in the final version.
>
> Thank you again for your suggestions. We hope to have addressed your concerns and kindly request that you consider increasing the score.

---

### Official Review · Reviewer_NQwV · 2024-03-24

**Q2-1 Originality-Novelty:** 4
**Q2-2 Correctness-Technical Quality:** 3
**Q2-5 Clarity Of Writing:** 3

**Q1 Summary And Contributions:**

The paper demonstrates a novel approach in integrating the Recursive Feature Machine (RFM) kernels with Gaussian Processes (GPs). The developed methods, named GP-RFM-Laplace and GP-RFM-diag, are endowed with the advantage of data-adaptive kernels without significant overhead. GP-RFM methods successfully improved uncertainty quantification performance of GPs, leading to competitive performance over boosting-based methods across datasets and metrics.

**Q2-3 Extent To Which Claims Are Supported By Evidence:**

4: Excellent: all claims are supported by very convincing evidence (in the form of comprehensive experimental evaluation, rigorous mathematical proofs, detailed (pseudo-)code, precise references, well-motivated and realistic assumptions) and the authors deliver what they promise.

**Q2-4 Reproducibility:**

2: Fair: key resources (e.g. proofs, code, data) are unavailable but key details (e.g. proof sketches, experimental setup) are sufficiently well-described for an expert to confidently reproduce the main results.

**Q3 Main Strengths:**

- The paper coherently presents a novel and elegant approach that bridges GP with RFM.
- Developed methods are able to demonstrate compelling results.
- Comprehensive evaluation settings and baselines.
- Extensive discussion on insights and limitations

**Q4 Main Weakness:**

The training and optimization procedures are not detailed.

**Q5 Detailed Comments To The Authors:**

- I suggest the author to elaborate more on the training and optimization procedures with pseudocode.
- Having GP on the title may help the paper with better exposures.

**Q9 Complying With Reviewing Instructions:**

Yes

---

> ### Author Rebuttal · Authors · 2024-04-05
>
> We thank the reviewer for the positive feedback and suggestions. We will add more explanation about the optimization procedure and add a pseudo code in the final version as it is suggested.

---

### Meta-Review · Area_Chair_4tYu · 2024-04-16

The focus of the manuscript is predictive modelling with uncertainty estimation and feature learning. In order to tackle this problem, the authors propose to use the Laplace kernel (7) (with an adjustable matrix parameter M) in Gaussian Processes (GPs), and learn M using the Average Gradient Outer Product method [Radhakrishnan et al., 2022]. The idea is illustrated numerically on multiple UCI and OpenML benchmarks. Though kernel methods and GPs are widely-used in data science (hence the focus is interesting), the contribution in the submission is somewhat moderate.